# An ancestral function of strigolactones as symbiotic rhizosphere signals

Kyoichi Kodama [1,13], Mélanie K. Rich[2,13], Akiyoshi Yoda [3,4,13], Shota Shimazaki [1,13], Xiaonan Xie[3,4,13], Kohki Akiyama [5], Yohei Mizuno[1], Aino Komatsu [1], Yi Luo[1], Hidemasa Suzuki [1], Hiromu Kameoka [1], Cyril Libourel [2], Jean Keller [2], Keiko Sakakibara [6], Tomoaki Nishiyama [7], Tomomi Nakagawa[8], Kiyoshi Mashiguchi [9], Kenichi Uchida[10], Kaori Yoneyama[11], Yoshikazu Tanaka[1], Shinjiro Yamaguchi [9], Masaki Shimamura[12], Pierre-Marc Delaux [2✉], Takahito Nomura [3,4✉] & Junko Kyozuka [1✉]

In flowering plants, strigolactones (SLs) have dual functions as hormones that regulate growth and development, and as rhizosphere signaling molecules that induce symbiosis with arbuscular mycorrhizal (AM) fungi. Here, we report the identification of bryosymbiol (BSB), an SL from the bryophyte *Marchantia paleacea*. BSB is also found in vascular plants, indicating its origin in the common ancestor of land plants. BSB synthesis is enhanced at AM symbiosis permissive conditions and BSB deficient mutants are impaired in AM symbiosis. In contrast, the absence of BSB synthesis has little effect on the growth and gene expression. We show that the introduction of the SL receptor of Arabidopsis renders *M. paleacea* cells BSB-responsive. These results suggest that BSB is not perceived by *M. paleacea* cells due to the lack of cognate SL receptors. We propose that SLs originated as AM symbiosis-inducing rhizosphere signaling molecules and were later recruited as plant hormone.

[1] Graduate School of Life Sciences, Tohoku University, Sendai, Japan. [2] LRSV, Université de Toulouse, CNRS, UPS, Toulouse INP, Auzeville-Tolosane, France. [3] United Graduate School of Agricultural Science, Tokyo University of Agriculture and Technology, Tokyo, Japan. [4] Center for Bioscience Research and Education, Utsunomiya University, Tochigi, Japan. [5] Graduate School of Life and Environmental Sciences, Osaka Prefecture University, Osaka, Japan. [6] Department of Life Science, Rikkyo University, Tokyo, Japan. [7] Research Center for Experimental Modeling of Human Disease, Kanazawa University, Kanazawa, Japan. [8] National Institute for Basic Biology, Okazaki, Japan. [9] Institute for Chemical Research, Kyoto University, Kyoto, Japan. [10] Department of Biosciences, Teikyo University, Tochigi, Japan. [11] Graduate School of Agriculture, Ehime University, Ehime, Japan. [12] Graduate School of Integrated Sciences for Life, Hiroshima University, Hiroshima, Japan. [13] These authors contributed equally: Kyoichi Kodama, Mélanie K. Rich, Akiyoshi Yoda, Shota Shimazaki, Xiaonan Xie. ✉email: pierre-marc.delaux@cnrs.fr; tnomura@cc.utsunomiya-u.ac.jp; junko.kyozuka.e4@tohoku.ac.jp

L and plants evolved from an algal ancestor more than 450 million years ago[1]. The first land plants faced many challenges, such as UV radiation and nutrient-poor soil. Thus, initial colonization of the terrestrial environment required the evolution of innovations such as the deployment of complex hormonal regulations and the mutualistic symbiosis formed with arbuscular mycorrhizal (AM) fungi, a monophyletic group of symbiotic soil-borne fungi[2–4]. During AM symbiosis, plants produce and supply carbohydrates and lipids to the fungus, in exchange for mineral nutrients, in particular phosphorus, mined in the soil by the fungal symbiont[5]. Symbiosis with AM fungi is observed in more than 80% of extant land plant species, including the bryophytes that diverged from the vascular plant lineage more than 400 million years ago[3]. In flowering plants, establishment of the AM symbiosis requires the activation of fungal metabolism and stimulation of hyphal branching by plant root-derived strigolactones (SLs)[6]. In addition to their role as rhizosphere signaling molecules, SLs also function as a class of plant hormones and regulate various aspects of growth and development in flowering plants[7,8]. The dual function, as hormones and as rhizosphere signaling molecules, makes SLs unique among plant hormones.

The function and signaling of SLs are well understood in flowering plants, whereas little is known outside this plant lineage[9,10]. The origin and evolution of their dual function are also largely unknown. One reason why SL research has been hampered is related to their nature. The chemical structure of SLs differs from species to species, many species have multiple SLs and they exist in cells in minute concentrations[11,12]. Phylogenetic studies and genome sequences revealed that genes in the SL biosynthesis pathway are conserved in bryophytes[13,14]. However, although initial studies reported SLs in some bryophytes, their actual presence is still under debate and, more importantly, their function remains unknown[12,14–16].

Here we report the identification of an ancestral SL, bryosymbiol (BSB), present in diverse bryophytes such as *Marchantia paleacea*, and vascular plants. In *M. paleacea*, BSB is secreted from the plants and is required for AM symbiosis, but not for development. We show that BSB is not perceived by *M. paleacea* cells due to the absence of cognate SL receptors. Our findings reveal that the ancestral function of SLs is as AM symbiosis-inducing rhizosphere signaling molecules and that this function was already present in the most recent common ancestor of land plants.

## Results

### Presence of SL biosynthesis genes is specific to AM symbiosis-forming species in *Marchantia*.
Among the three extant groups of bryophytes, AM symbiosis is observed in liverworts and hornworts, but not in mosses[2,17]. In liverworts, hyphae of AM fungi enter through rhizoids and produce arbuscules in parenchymatous cells in the midrib region of the thallus (Fig. 1a). We confirmed that AM symbiosis is widely observed in the genus *Marchantia*, including *M. paleacea*, *M. emarginata* and *M. pinnata*, whereas it is absent in *M. polymorpha*, the model liverwort species used for molecular genetic studies (Fig. 1a)[17,18].

To date, more than 30 SLs have been identified in root exudates of various plant species[12]. SL biosynthesis starts from the conversion of all-*trans*-β-carotene to 9-*cis*-β-carotene by DWARF27 (D27)[19] (Fig. 1b). Then, 9-*cis*-β-carotene is converted to carlactone (CL) by the consecutive actions of two carotenoid cleavage dioxygenases (CCD7 and CCD8)[20–22]. CL is converted to carlactonoic acid (CLA), the universal precursor of a variety of species-dependent SLs, by the cytochrome P450, CYP711A, encoded by *MORE AXILLARY GROWTH 1* (*MAX1*) homologs[23].

CLA is converted to canonical SLs such as 4-deoxyorobanchol (4DO), orobanchol and 5-deoxystrigol by species-specific functions of the P450s, CYP711A and CYP722C[24,25]. The *M. paleacea* genome contains two orthologs of *D27*, one of *CCD7*, two of *CCD8* and one of *MAX1*[17] (Supplementary Figs. 1 and 2 and Supplementary Data 1). There are no CYP722 family genes in any of the investigated *Marchantia* species[17,26]. *CCD8* and *MAX1* orthologs exist in all other AM symbiosis-forming *Marchantia* species tested (Supplementary Figs. 1 and 2). Despite the polymorphisms present in the genome sequences, the overall structure of the genes is well conserved and the amino acid sequences in the conserved domains show high similarities (Fig. 1c). In contrast to this, both *CCD8* and *MAX1* orthologs are absent from the genome of *M. polymorpha*, a non-AM symbiosis-forming species, indicative of gene loss[26]. Together, these phylogenetic analyses suggest that AM symbiosis-forming *Marchantia* species produce SLs.

### Bryosymbiol (BSB), a previously unidentified SL is produced by *M. paleacea*.
The correlation between the presence of the *CCD8* and *MAX1* genes and the occurrence of AM symbiosis suggested the possibility that AM symbiosis-forming *Marchantia* species produce SLs. Using LC-MS/MS analysis, we detected CLA (retention time of 11.86 min) and its unknown analog (retention time of 10.73 min) in the rhizoid exudates of *M. paleacea* (Fig. 2a, b). Next, to determine whether the CLA in *M. paleacea* is synthesized through the same pathway as in flowering plants, we generated knock-out mutants of the two *CCD8* genes (named Mpa*ccd8a* Mpa*ccd8b*, or Mpa*ccd8a/8b*) and one *MAX1* (Mpa*max1*) gene using CRISPR/Cas9 (Supplementary Table 1). CLA was not detected in either Mpa*ccd8a/8b* or Mpa*max1* mutants, indicating that CLA biosynthesis is dependent on both MpaCCD8 and MpaMAX1 (Fig. 2a).

To test for the presence of downstream SL activity, we used a parasitic seed germination assay. This is based on the ability of obligate root parasitic Orobanchaceae plants, such as witchweed (*Striga* spp.) and broomrape (*Orobanche* and *Phelipanche* spp.), to use SLs as host recognition signals in the rhizosphere[27]. This assay is sensitive to concentrations of SLs as low as 10 pM. We fractionated the rhizoid exudates of *M. paleacea* by reversed-phase HPLC and found germination stimulation activity in the fraction collected at around 8 min for all three species of parasitic plants (Fig. 2c). Since there is no known SL eluting at this retention time[28], we hypothesized that *M. paleacea* produces an unidentified SL. To identify the nature of this compound, we collected and purified the active fraction from the rhizoid exudate of *M. paleacea* in a large-scale culture. The structure was determined as (*R*)-5-((*E*)-((4*S*,5*R*)-4-hydroxy-2-oxo-5-(2,6,6-trimethylcyclohex-1-en-1-yl)dihydrofuran-3(2*H*)-ylidene)methoxy)-3-methylfuran-2(5*H*)-one by an analysis of 1D/2D NMR and HR-ESI- and EI-MS data, statistical DP4+ analyses based on DFT-GIAO NMR calculations, quantum chemistry ECD calculations (Supplementary Figs. 3–8, Supplementary Table 2 and Supplementary Data 2). We named this newly identified SL BSB. To compare with SL, we adopted the classic SL numbering of atoms for BSB as shown in Fig. 2d.

BSB was detected in the exudate and plant extracts of WT *M. paleacea* but not in the Mpa*ccd8a/8b* and Mpa*max1* mutants (Figs. 2e, f and 3a). In contrast, we detected a higher level of CL accumulation in the extract of the Mpa*max1* mutant than that of the WT (Fig. 3b). This suggests that the substrate of MpaMAX1 is likely to be CL, similar to that of MAX1 homologs in vascular plants. To determine whether MpaMAX1 directly converts CL to BSB, we performed in vitro metabolic experiments. When the microsomal fraction of yeast expressing Mpa*MAX1* was incubated with CL, the production of CLA and BSB were detected

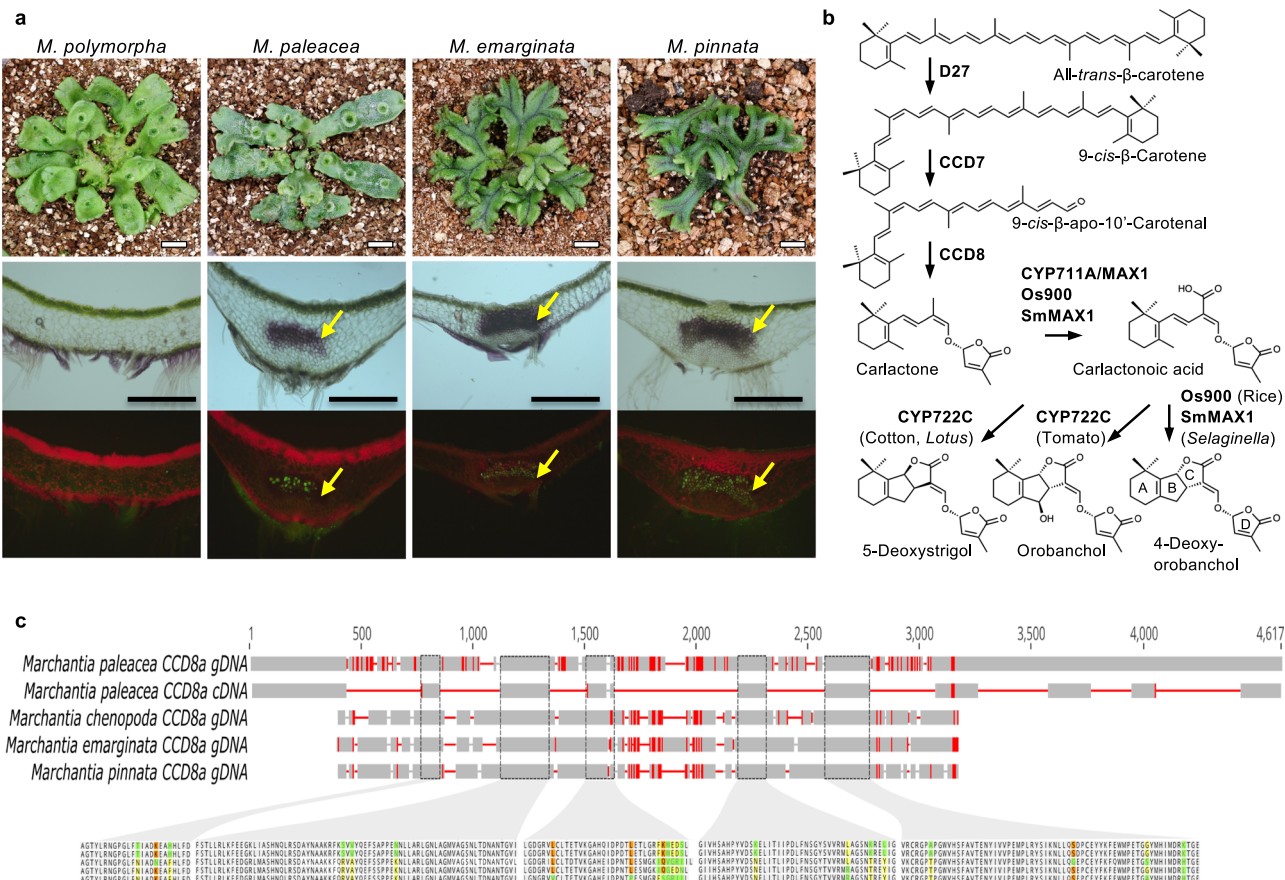

**Fig. 1 Presence of SL biosynthesis genes is specific to AM symbiosis-forming species in *Marchantia*. a** *Marchantia* species, namely *Marchantia polymorpha*, *Marchantia paleacea*, *Marchantia emarginata* and *Marchantia pinnata*, used in this study. Upper panels show plant morphology. Scale bars: 1 cm. Middle and lower panels are transverse sections after colonization by *Rhizophagus irregularis*. The lower panel shows WGA-FITC staining of the fungi. Arbuscules are formed in the midrib region in the thallus (arrows). Experiment was repeated three times with similar results. Scale bars: 0.5 mm. **b** Biosynthesis pathway for strigolactones in seed plants and lycophytes. **c** Conservation of the *CCD8A* genes in mycorrhizal Marchantia species. Amplified sequences of *CCD8A* are aligned against the *Marchantia paleacea* gene model. Red lines indicate polymorphisms. The predicted amino acid sequences in the conserved domains are shown below the gene model. Polymorphic amino acids are colored. Equivalent amino acids present in one/two, three or four sequences are colored in green, yellow and orange respectively.

(Fig. 3c). This indicates that MpaMAX1 catalyzes two steps, namely the conversion from CL to CLA and CLA to BSB. To further confirm this pathway, we incubated microsomes of yeast expressing Mpa*MAX1* with CLA and detected BSB (Fig. 3c). We concluded that MpaMAX1 produces BSB from CL via CLA. The conversion of CLA to BSB appears to proceed via 7,8-epoxidation (CL numbering) followed by $S_N2$-type ring opening of the epoxide by the carboxyl oxygen atom to yield the possible two isomers with 4,5-*anti*-substituted butyrolactone ring (Fig. 3d). Stereoselective closure of the C ring may occur, producing BSB (4*S*,5*R*,2′*R*) and its diastereomer (4*R*,5*S*,2′*R*), probably depending on the epoxy configuration. However, the occurrence of 7,8-epoxy-CLA, a putative intermediate between CLA and BSB (Fig. 3d), could not be confirmed due to the lack of the authentic standard.

To determine whether BSB is a derived trait of *M. paleacea* or more widespread across the plant diversity, we investigated its presence in diverse land plants. First, we analyzed *M. pinnata*, *M. emarginata* and *M. polymorpha*. CLA and BSB were detected in exudates from the two AM symbiosis-forming *Marchantia* species but not from *M. polymorpha*, consistent with the presence/absence of *CCD8* and *MAX1* (Fig. 2a, e, f and Supplementary Figs. 1 and 2). Neither CLA nor BSB was detected in the extracts of the model moss *Physcomitrium patens*, which

lacks the *MAX1* gene (Fig. 2a, e)[29]. Since *CCD8* and *MAX1* are present in hornworts[30,31], we analyzed exudates from the hornwort *Phaeoceros carolinianus*. We also analyzed nine ferns that have been obtained from a garden store and 27 seed plants, including plants in which SLs have not been detected or analyzed so far (Supplementary Table 3). CLA and BSB (retention time of 8.3 min), and/or a likely stereoisomer of BSB (retention time of 7.7 min), were detected in exudates from the hornwort *P. carolinianus*, the ferns *Dryopteris erythrosora* and *Matteuccia struthiopteris*, and the seed plants *Asparagus officinalis* (monocot) and *Arachis hypogaea* (dicot) (Fig. 2a, e and Supplementary Fig. 9).

Collectively, these results demonstrate that BSB is produced by AM symbiosis-forming *Marchantia* and a variety of vascular plants. Because bryophytes, such as liverworts and hornworts, and vascular plants, such as ferns and flowering plants, diverged more than 400 million years ago, it is inferred that BSB was likely present in the most recent common ancestor of land plants.

**Enhanced BSB synthesis correlates with an AM symbiosis permissive state.** In flowering plants, phosphate starvation stimulates the expression of SL-biosynthesis genes and the actual production of SLs, and correlates with a permissive state for the

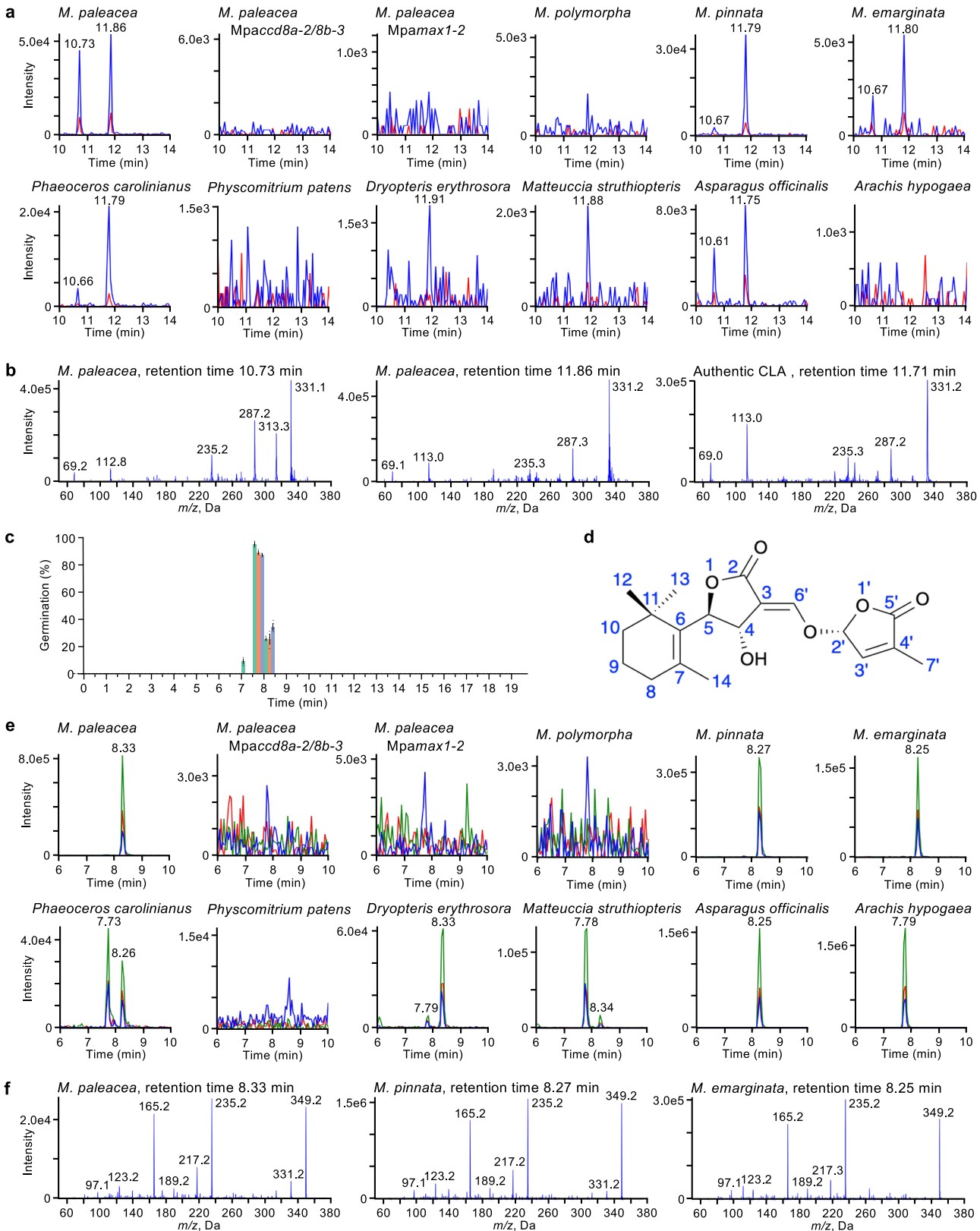

AM symbiosis to be established[7,32]. We tested whether BSB synthesis is also enhanced in *M. paleacea* under phosphate-deficient conditions. The relative transcript levels of all five genes in the SL synthesis pathway, namely, Mpa*D27*, Mpa*CCD7*, Mpa*CCD8A*, Mpa*CCD8B* and Mpa*MAX1*, were increased upon phosphate starvation (Fig. 4a). We also found that the relative

transcript level of the *CCD8* gene in *Anthoceros agrestis*, a hornwort, was induced by phosphate starvation (Fig. 4a).

To obtain a genome-wide view of the phosphate level-dependent gene expression in *M. paleacea*, we performed RNA-seq analysis (Fig. 4b). The expression of many genes changed under phosphate-deficient conditions (Supplementary Data 3).

**Fig. 2 Bryosymbiol (BSB), a previously unidentified SL is produced by *M. paleacea*. a** Detection of carlactonoic acid. Carlactonoic acid in the exudates of different *Marchantia* species and the hornwort *Phaeoceros carolinianus*, protonema extract of the moss *Physcomitrium patens*, root exudates of ferns *Dryopteris erythrosora* and *Matteuccia struthiopteris*, and seed plants *Asparagus officinalis* and *Arachis hypogaea* were analyzed by LC-MS/MS. Multiple reaction monitoring (MRM) chromatograms of carlactonoic acid (blue, 331.10/113.00; red, 331.10/69.00; *m/z* in negative mode) by LC-MS/MS are shown. **b** Identification of carlactonoic acid. Product ion spectra derived from the precursor ion (*m/z* 331 in negative mode) of the peaks detected in the exudates of *M. paleacea* and authentic carlactonoic acid (CLA) are shown. **c** Germination stimulation activity of the exudate of *M. paleacea* on root parasitic plants. The exudate was separated by reversed-phase HPLC every 30 s. All the fractions were tested for seed germination activity on *Orobanche minor* (green), *Striga hermonthica* (orange) and *Phelipanche ramosa* (blue). Data are means ± SE (∼30 seeds per disk, *n* = 3). The seed germination activity (%) of *rac*-GR24 ($10^{-8}$ M) was 99.4 ± 0.6 on *O. minor*, 69.2 ± 4.2 on *S. hermonthica* and 38.7 ± 1.7 on *P. ramosa*. **d** Chemical structure of bryosymbiol (BSB). Numbering of atoms for BSB was adopted as the classic SLs. **e** Detection of BSB. BSB in the exudates of *Marchantia* species and the hornwort *P. carolinianus*, protonema extract of the moss *P. patens*, root exudates of ferns *D. erythrosora* and *M. struthiopteris*, and seed plants *A. officinalis* and *A. hypogaea* were analyzed by LC-MS/MS. MRM chromatograms of BSB (blue, 349.00/97.00; red, 349.00/165.00; green, 349.00/235.00; *m/z* in positive mode) are shown. **f** Product ion spectra of BSB. Product ion spectra derived from the precursor ion (*m/z* 349 in positive mode) of peaks detected in *Marchantia* species are shown.

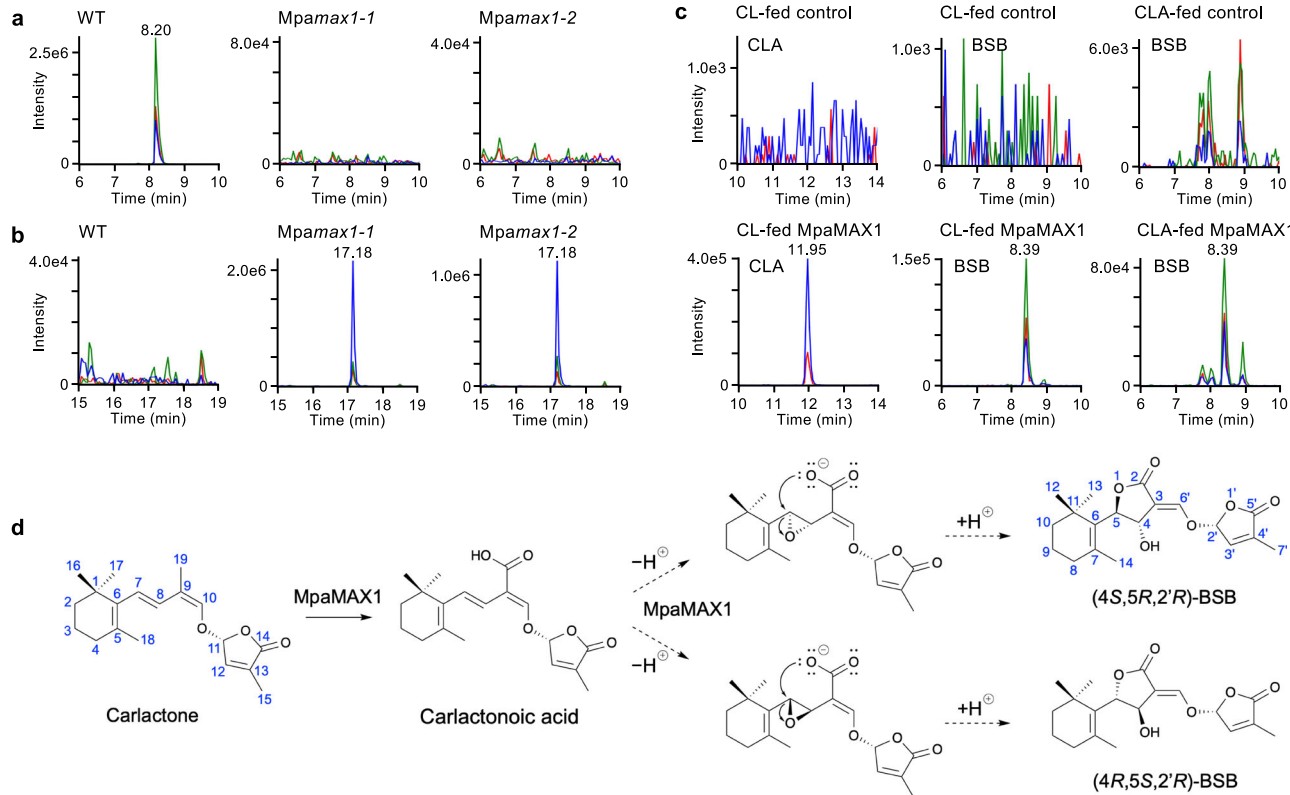

**Fig. 3 MpaMAX1 produces BSB from carlactone via carlactonoic acid. a**, **b** Detection of BSB and carlactone in plant tissues of *Marchantia*. Whole plants of WT and the Mpa*max1* mutant (two alleles) were extracted and analyzed by LC-MS/MS. MRM chromatograms of BSB (**a**) and carlactone (**b**) (blue, 303.00/97.00; red, 303.00/189.00; green, 303.00/207.00; *m/z* in positive mode) are shown. **c** Detection of carlactonoic acid and BSB in feeding experiments with recombinant MpaMAX1. *rac*-Carlactone (CL) or *rac*-carlactonoic acid (CLA) was incubated with the microsomes of MpaMAX1-expressed yeast. Yeast microsomes with an empty vector were used as a control. MRM chromatograms of carlactonoic acid and BSB in extracts of the microsomes are shown. **d** Proposed mechanism of the conversion to BSB from carlactone. MpaMAX1 oxidizes C-19 to produce carlactonoic acid and catalyzes the α- or β-epoxidation of Δ7,8, followed by $S_N2$-type ring opening of the epoxide by the carboxyl oxygen atom to yield the possible two isomers with 4,5-anti-substituted butyrolactone ring.

In petunia and *Medicago truncatula*, expression of *PDR1*, an SL transporter gene, is also enhanced under low phosphate conditions, implying that increased amounts of SLs are exuded to the rhizosphere under phosphate starvation[33,34]. Although SL transporters in bryophytes have not yet been identified, ABCG transporter genes, homologs of *PDR1*, were up-regulated under phosphate-deficient conditions in *M. paleacea* (Fig. 4c).

We then tested the level of exuded BSB using the parasitic seed germination assay under low phosphate conditions. Consistent with the enhanced expression of SL biosynthesis and putative transporter genes, seed germination frequency was higher in exudates of *M. paleacea* grown under phosphate-deficient

conditions (Fig. 4d), implying that the amount of BSB exuded was increased upon phosphate starvation. We finally examined if phosphate starvation affects AM symbiosis in *M. paleacea* as in vascular plants (Fig. 4e). While phosphate-deficient conditions were permissive for AM symbiosis, increasing level of phosphate in the watering solution resulted in a gradual decrease in colonization by the AM fungus, illustrated by an increased zone without AM symbiosis (Fig. 4e).

AM symbiosis does not occur during the early stages of *M. paleacea* development and is first observed in 3-week-old gemmalings. Thus, we also tested the developmental control of BSB synthesis. We found enhanced relative transcript level of

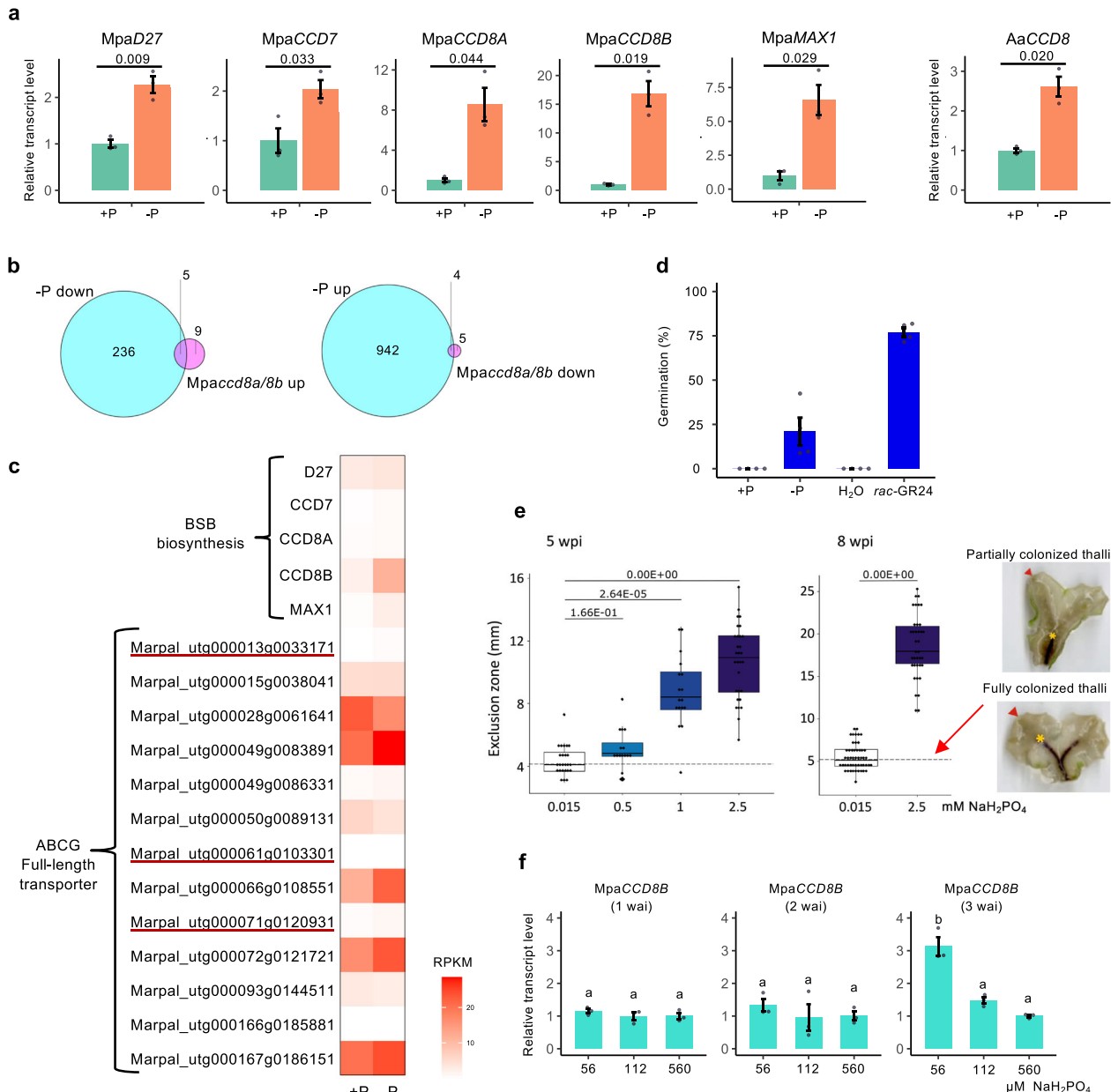

**Fig. 4 Enhanced BSB synthesis correlates with an AM symbiosis permissive state. a** Increased transcript level of the SL biosynthesis genes, Mpa*D27*, Mpa*CCD7*, Mpa*CCD8A*, Mpa*CCD8B* and Mpa*MAX1*, of *M. paleacea*, and Aa*CCD8* of *Anthoceros agrestis* under phosphate starvation. Plants were grown on half-strength Gamborg's B5 medium with (+P) or without (−P) phosphate. Data are means ± SD (*n* = 3 biologically independent samples) and *p* values in Welch's *t*-test, two tailed are indicated. **b** Identification of DEGs under phosphate starvation and in Mpa*ccd8a/8b* mutants by RNA-seq analysis. Left: Venn diagram of genes down-regulated in −P condition compared to +P condition in WT (blue) and up-regulated in Mpa*ccd8a/8b* compared to WT (pink). Right: Venn diagram of genes up-regulated in −P compared to +P condition in WT (blue) and down-regulated in Mpa*ccd8a/8b* compared to WT (pink). **c** Heatmap of gene transcript level under phosphate deficiency in *M. paleacea*. DEGs (logFC > 1) up-regulated under –P are underlined. **d** Increase in the stimulation activity of the exudate of *M. paleacea* on seed germination of *Orobanche minor* under phosphate starvation. Exudates from *M. paleacea* grown on +P and −P medium were used for analysis. Data are means ± SD (*n* = 4 biologically independent samples). **e** Low phosphate allows mycorrhizal colonization. *M. paleacea* inoculated with *Rhizophagus irregularis* were watered with long ashton solutions containing 0.015, 0.5, 1 or 2.5 mM phosphate. Right: Cleared thalli of *M. paleacea* grown in low phosphate (bottom right) show a higher level of fungal colonization (marked by a natural purple pigment) than thalli grown at high phosphate (upper right). The exclusion zone which is the distance between the notch (harrow) and the end of the colonized zone (yellow asterisk) was measured (*n* ≥ 15 biologically independent plants). Box plot shows first quartile, median, third quartile, whiskers 1.5 interquartile. *p* values (one-way ANOVA, post hoc Tukey) are shown. **f** Developmental stage-dependent induction of Mpa*CCD8B* transcript level in response to low phosphate conditions. Gemmalings of 1–3 weeks after inoculation (wai) were transferred to media containing different concentrations of phosphate. Data are means ± SD (*n* = 3 biologically independent samples) and the HSD test was used for multiple comparisons. Statistical differences (*p* values < 0.05) are indicated by different letters.

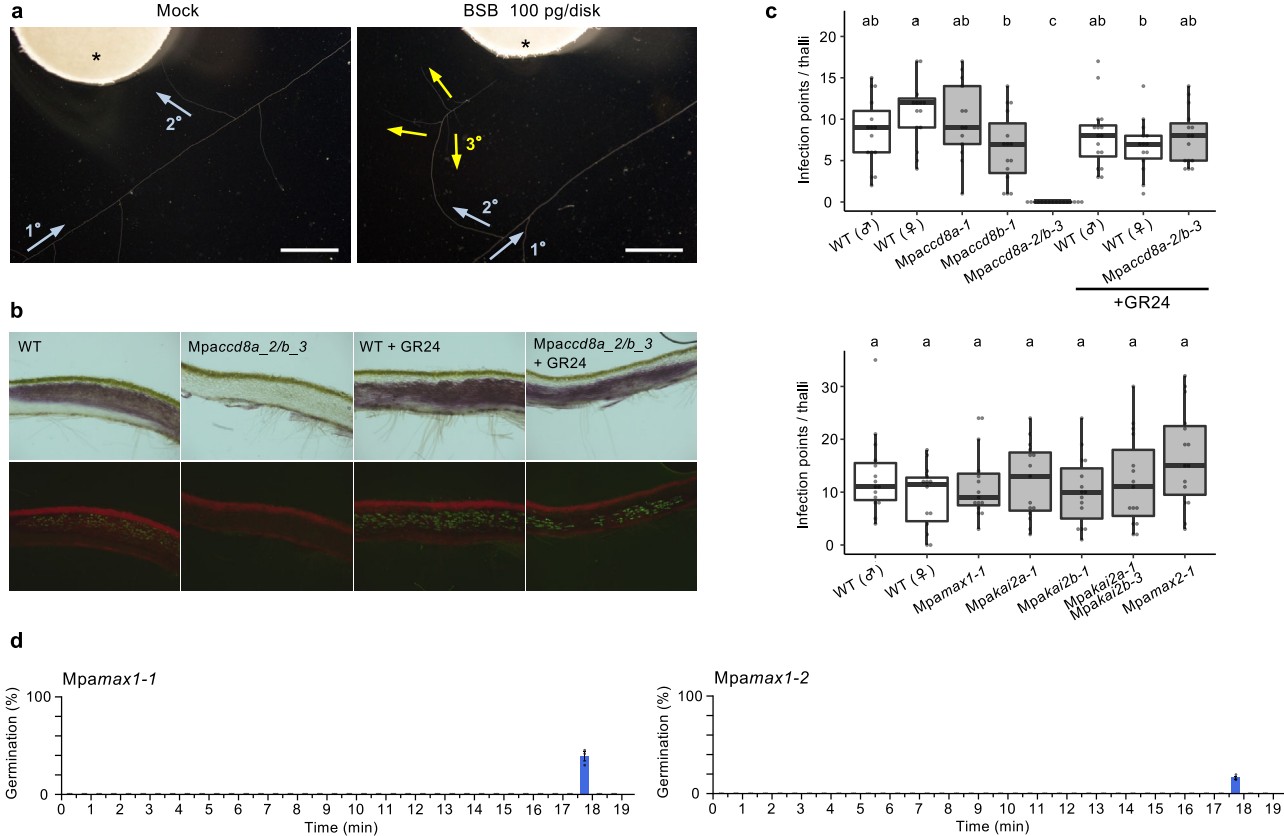

**Fig. 5 BSB is required for symbiosis with AM fungi in M. paleacea. a** Hyphal branching of *Gigaspora margarita* induced by BSB (100 pg per disk) using the paper disk diffusion method. Arrows indicate the direction of growth of primary, secondary and tertiary hyphae. Asterisks indicate paper disks. Experiment was repeated three times with similar results. **b** Upper and bottom panels are transverse sections of colonization by *Rhizophagus irregularis*. The bottom panels show WGA-FITC staining of the fungi. Experiment was repeated three times with similar results. Scale bars: 0.5 mm. **c** The number of infection points per thalli ($n \geq 13$ biologically independent plants). Box plot shows first quartile, median, third quartile, whiskers 1.5 interquartile. Letters show different statistical groups ($p$ values < 0.05, one-way ANOVA, post hoc Tukey). **d** Germination stimulation activity on root parasitic plants of the exudates of the Mpa*max1* mutant. The exudates of the two alleles were separated by reversed-phase HPLC. Every 30 s all the fractions were tested for seed germination activity on *Orobanche minor*. Data are means ± SE (~30 seeds per disk, $n = 3$). The seed germination activity (%) of *rac*-GR24 ($10^{-8}$ M) was 99.4 ± 0.6 on *O. minor*.

Mpa*CCD8B* under low phosphate conditions in 3-week-old gemmalings but not in earlier stage gemmalings (Fig. 4f and Supplementary Fig. 10). Altogether, these results indicate that BSB synthesis correlates with a permissive state for AM symbiosis in *M. paleacea*.

**BSB is required for symbiosis with AM fungi in M. paleacea.** The correlation between the presence of BSB and AM symbiosis suggests that bryophytes use BSB as a rhizosphere signaling chemical to promote the establishment of symbiosis with AM fungi. We confirmed that BSB induces hyphal branching in the AM fungus *Gigaspora margarita* with a minimum effective concentration of 10 pg/disk, which is within the range of the effects of previously described natural SLs (1 to 100 pg/disk) (Fig. 5a)[35]. To further test the symbiotic function of BSB in vivo, the Mpa*ccd8a*, Mpa*ccd8b* and Mpa*ccd8a/8b* mutants were inoculated with spores of the AM fungus *Rhizophagus irregularis*. Six weeks after inoculation, the Mpa*ccd8a* and Mpa*ccd8b* mutant thalli showed colonization similar to the WT. In contrast, almost all thalli of the Mpa*ccd8a/8b* mutants were not colonized (Fig. 5b, c and Supplementary Fig. 11). To determine which stages of the symbiotic establishment were affected in the mutants, the number of infection points was scored. While the number of infection points was similar in the WT plants and the Mpa*ccd8a* and Mpa*ccd8b* single mutants, both alleles of the Mpa*ccd8a/8b* mutant showed

no or very few infection sites (Fig. 5c and Supplementary Fig. 11). To further determine whether the lack of infection was due to the absence of stimulation of hyphal branching by BSB, complementation assays were conducted using *rac*-GR24, a synthetic analog of SL known to induce hyphal branching. Exogenous treatment with 100 nM *rac*-GR24 restored colonization in the two Mpa*ccd8a/8b* mutant alleles, although the total number of infection points was highly variable (Fig. 5b, c and Supplementary Fig. 11).

In flowering plants, D14LIKE (D14L), an ortholog of KARRIKIN INSENSITIVE2 (KAI2), and MAX2-dependent signaling positively regulate AM symbiosis[36,37]. In contrast to this, AM symbiosis was not affected in *kai2* or *max2* mutants of *M. paleacea* (Fig. 5c and Supplementary Fig. 11). This suggests that KAI2 and MAX2-dependent signaling were recruited to control AM symbiosis in vascular plant lineages.

In contrast to the Mpa*ccd8a/8b* mutants, neither colonization nor infection sites were affected in Mpa*max1* mutants, despite the absence of BSB production (Fig. 5c and Supplementary Fig. 11). To understand this discrepancy, we pursued the presence of an active SL other than BSB in the exudates of Mpa*max1* mutants. We fractionated the exudates from Mpa*max1* by reversed-phase HPLC and tested for the presence of SL-like activity using the parasitic seed germination assay. These assays identified a single active fraction at the retention time of 18 min (Fig. 5d),

while there was no activity in the corresponding fraction of WT exudates (Fig. 2c). This fraction contains CL (Fig. 3b), which is known to induce hyphal branching, although at higher concentrations than BSB[38]. Accumulation of CL in the Mpa*max1* mutants may explain the observed normal fungal colonization.

From this reverse genetic approach and chemical complementation assays, we conclude that the function of SLs as symbiotic signal in the rhizosphere is an ancestral trait in land plants, which has been maintained in both the vascular plants and the Marchantia for 400 million years.

**BSB-deficient mutants of *M. paleacea* show no abnormal developmental patterns and gene expression profiles.** We next examined the possible endogenous functions of BSB in *M. paleacea*. Despite the absence of BSB synthesis, no alteration in growth pattern or morphology was observed in Mpa*ccd8a/8b* and Mpa*max1* loss-of-function mutants throughout their growth, even in plants grown under the low phosphate conditions that stimulate BSB synthesis (Fig. 6a–d). This is consistent with the results of the RNA-seq analysis comparing gene expression profiles between WT and Mpa*ccd8a/8b*, which shows that only a small number of genes are differentially regulated (Fig. 4b). These data suggest that the absence of endogenous BSB causes little effect on *M. paleacea*. Feedback regulation of SL synthesis genes such as *CCD8* is observed in many seed plants[7,39], whereas relative transcript levels of Mpa*CCD8A* and Mpa*CCD8B* genes are not affected by exogenously applied SLs (Fig. 6e).

In flowering plants, SL is perceived by D14, encoding an α/β-hydrolase protein[40–42]. D14 originated from a gene duplication of *KAI2* before the radiation of seed plants[43]. Although endogenous ligands of KAI2 are unknown, it is thought that D14 and KAI2 interact with other signaling components such as the F-box protein MAX2 upon the perception of their cognate ligands[9]. This leads to the degradation of SMXL proteins that work as regulators of transcription of downstream genes[44,45]. We previously reported that KAI2, MAX2 and SMXL function in a common proteolysis-dependent signaling pathway in *M. polymorpha*[46]. Similar to *M. polymorpha*, the *M. paleacea* genome contains one *MAX2* (Mpa*MAX2*) and two *KAI2* genes (Mpa*KAI2A* and Mpa*KAI2B*) but no *bona fide D14*, as is the case in other non-seed plants. To pursue a possibility that BSB plays an endogenous function in *M. paleacea*, we tested the possibility that MpaKAI2A and/or MpaKAI2B act as the receptors of BSB. Mpa*kai2a* and Mpa*max2* mutants (Supplementary Table 1) showed altered growth patterns, including reduced and upward growth of the early-stage thalli and delayed gemma cup initiation (Fig. 6a, b). Such developmental defects were not observed in the Mpa*ccd8a/8b* or the Mpa*max1* mutants (Fig. 6a, b). To further investigate the potential link between MpaKAI2A and MpaKAI2B and SL, their physical interaction with four stereoisomers of the synthetic SL GR24 was investigated by differential scanning fluorimetry (DSF) analysis (Supplementary Fig. 12a, b). Only (−)-GR24, which has an unnatural D-ring configuration and has been shown to interact with Arabidopsis KAI2[47], was able to weakly interact with MpaKAI2A whereas no sign of interaction was observed between MpaKAI2B and GR24. In flowering plants, SL synthesis is enhanced in SL signaling mutants because of the loss of feedback regulation[48]. However, no such increase was observed in the levels of CLA and BSB in exudates of Mpa*kai2a/2b* and Mpa*max2* mutants (Supplementary Fig. 12c–f). Altogether, these phenotypic and biochemical analyses indicate that MpaKAI2A, MpaKAI2B and MpaMAX2 are not involved in the perception of BSB or other metabolites downstream CCD8 or MAX1 in *M. paleacea*.

**Heterologous expression of AtD14, the SL receptor of Arabidopsis, renders *M. paleacea* cells BSB-sensitive.** From the

previous phylogenetic studies and the experimental work presented above, it can be proposed that BSB does not function as a hormone in Marchantia due to the absence of cognate receptors. In other words, although an SL, BSB, is produced and the known signaling pathway is present in Marchantia, the lack of the upstream receptor results in the two pathways being disconnected. If this hypothesis is valid, mimicking the seed plant duplication of the KAI2 homologs by adding a cognate SL receptor in *M. polymorpha* and *M. paleacea* should lead to SL responsiveness. To experimentally test this hypothesis, we took a gain-of-function approach in which the canonical SL receptor D14 of Arabidopsis (AtD14) was fused with the promoter of *M. polymorpha Elongation Factor1* (*EF1*) gene and introduced into *M. polymorpha* and *M. paleacea*. We examined whether endogenous BSB is recognized by the introduced AtD14, leading to signal transduction. More than 20 independent lines in each species were produced and two lines that show high and low relative transcript levels of At*D14* were selected from each species (named Mp*AtD14*ox and Mpa*AtD14*ox, for *M. polymorpha* and *M. paleacea*, respectively) for further analysis (Fig. 7a and Supplementary Fig. 13a). First, we tested if the introduction of AtD14 enables SL perception and signal transduction in response to *rac*-GR24 in *M. polymorpha* cells that do not produce BSB. Previously, we performed transcriptome analysis comparing WT, Mp*kai2a*, Mp*kai2b*, and Mp*max2* mutants. We identified many genes differentially expressed in the mutants, including a putative *Dienelactone Hydrolase Like Protein1* (*DLP1*) and Mp*SMXL* genes, whose expression is down-regulated in the mutants[46,49]. Induction of Mp*DLP1* transcription by addition of (−)-GR24 in WT *M. polymorpha* was also reported, however function of the MpDLP1 is unknown[49]. We used Mp*DLP1* and Mp*SMXL* genes as markers. The relative transcript levels of Mp*DLP1* and Mp*SMXL* were enhanced by the application of *rac*-GR24 in the Mp*AtD14*ox lines but not in WT plants (Supplementary Fig. 13b). To verify that AtD14 functions in the same signaling pathway as the endogenous KAI2A, Mp*kai2a* mutation was introduced into the Mp*AtD14*ox lines. By contrast with the Mp*kai2a* mutants, the upward thallus growth phenotype was rescued by application of (−)-2′-*epi*-GR24 and (+)-GR24, which have a 2′(*R*) configuration as natural SLs[47], in the Mp*AtD14*ox/Mp*kai2a* lines (Supplementary Fig. 13c).

Phenotypes of Mpa*kai2a*, the loss-of-function mutant of *KAI2A* in *M. paleacea*, closely resembled those of Mp*kai2a* (Figs. 6b and 7b). As for *M. polymorpha*, the thallus upward curving was rescued by application of (−)-2′-*epi*-GR24 and (+)-GR24 in the Mpa*AtD14*ox/Mpa*kai2a* (Fig. 7b). These results indicate that introducing AtD14 enables SLs perception and downstream signal transduction through the MAX2-dependent signaling pathway in both *M. paleacea* and *M. polymorpha*.

We next directly tested the effect of exogenously applied SLs on the signaling cascade. As is the case in *M. polymorpha*, neither (−)-2′-*epi*-GR24 nor (+)-GR24 enhanced the relative transcript levels of the Mpa*DLP1* and Mpa*SMXL* genes in WT *M. paleacea* plants, even at the highest concentration (Fig. 7c, d). This implies that the applied SLs are not perceived by WT *M. paleacea* cells, consistent with the disconnection between SL and the MAX2-dependent signaling pathway. In contrast, in the Mpa*AtD14*ox lines containing the SL receptor gene, the relative transcript levels of Mpa*DLP1* and Mpa*SMXL* were enhanced by the application of either (−)-2′-*epi*-GR24 or (+)-GR24 in a concentration-dependent manner (Fig. 7c, d).

We showed that BSB synthesis is enhanced under phosphate starvation (Fig. 4a, d). Therefore, we hypothesized that enhanced synthesis of BSB may enhance MAX2-dependent signaling under phosphate-deficient conditions. Indeed, the relative transcript levels of Mpa*DLP1* and Mpa*SMXL* were induced in Mpa*AtD14*ox lines

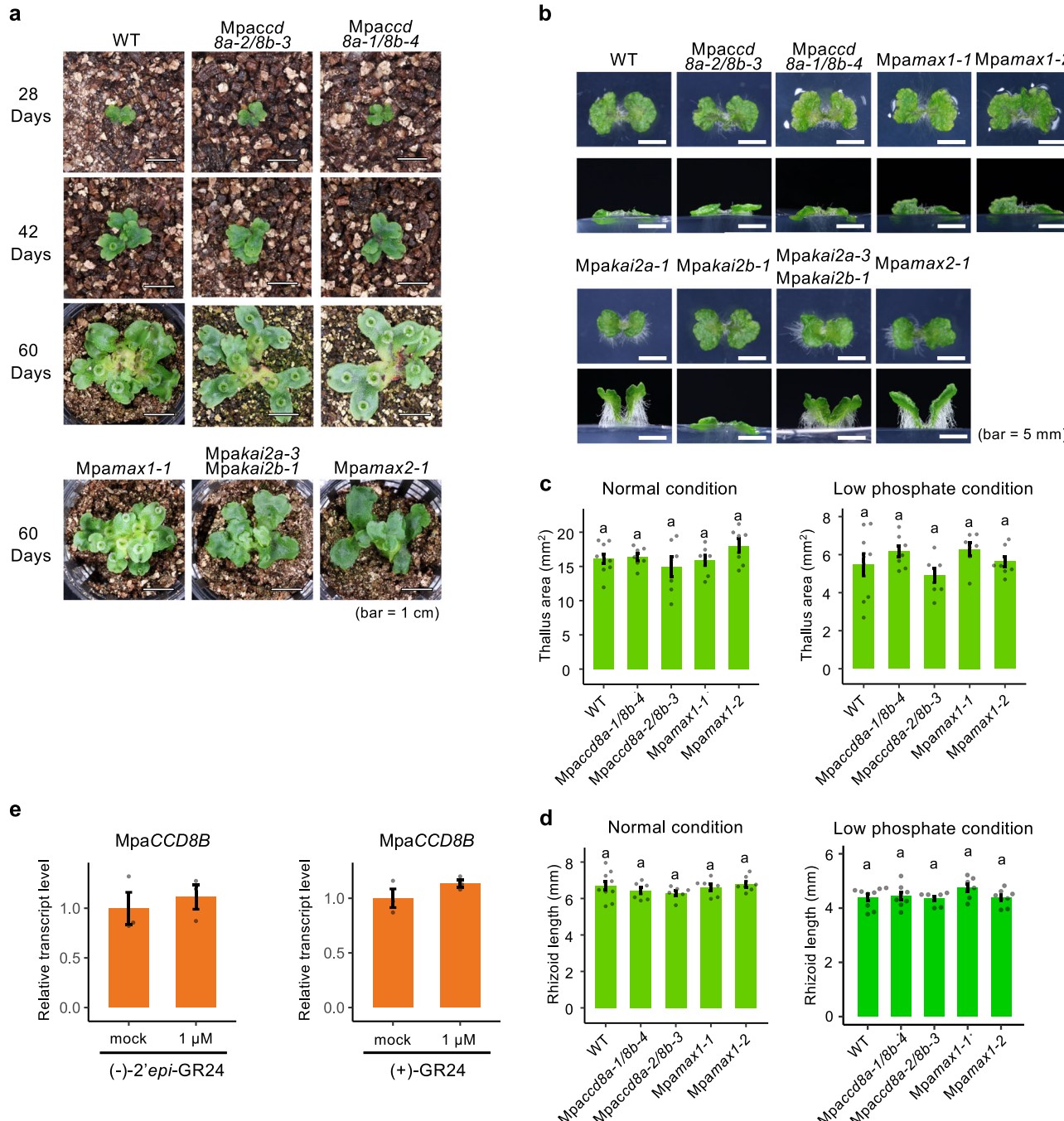

**Fig. 6 BSB-deficient mutants of _M. paleacea_ show no abnormal developmental patterns and gene transcript level profiles.** Growth of BSB-deficient (Mpa*ccd8a-2/8b-3*, Mpa*ccd8a-1/8b-4,* Mpa*max1-1* and Mpa*max1-2*) and KAI2 signaling (Mpa*kai2a-1*, Mpa*kai2b-1*, Mpa*kai2a-3* Mpa*kai2b-1* and Mpa*max2-1*) mutants of _M. paleacea_ grown in soil for 28, 42 and 60 days (**a**) and in media for 20 days (**b**). Thallus area and rhizoid length of BSB-deficient mutants of _M. paleacea_ grown on half-strength Gamborg's B5 medium containing 560 μM (**c**) and 56 μM (**d**) phosphorus (NaH$_2$PO$_4$) for 14 days. Data in **c**, **d** are means ± SD ($n \geq 7$ biologically independent plants) and the HSD test was used for multiple comparisons. Statistical differences (_p_ values <0.05) are indicated by different letters. **e** Transcript level of Mpa*CCD8B* in WT plants after (−)-2′-*epi*-GR24 and (+)-GR24 treatments. Data are means ± SD ($n = 3$ biologically independent samples).

under phosphate starvation (Fig. 7e). Although a slight increase in the relative transcript level of _SMXL_ under phosphate depletion was sometimes observed in WT plants, the same trend was detected in Mpa*ccd8a/8b*, suggesting that the slight increase was caused by unknown SL-independent mechanisms (Supplementary Fig. 13d). Besides the activation of marker genes, we tested whether the gain of the cognate SL receptor was sufficient to recruit BSB as an endogenous signal with a physiological effect. Thus, Mpa*kai2a* and

Mpa*AtD14*ox/Mpa*kai2a* lines were transferred to a medium with a lower level of phosphate. On this medium, while the upward curving phenotype of the Mpa*kai2a* lines remained fully unaffected, this defect was partially, but significantly, rescued in the Mpa*AtD14*ox/Mpa*kai2a* lines (Fig. 7f). Taken together, these results indicate that endogenous BSB can be perceived by a newly integrated SL receptor and transduce the signal through MAX2-dependent signaling pathway; that is, BSB is perceived in _M. paleacea_ cells when the

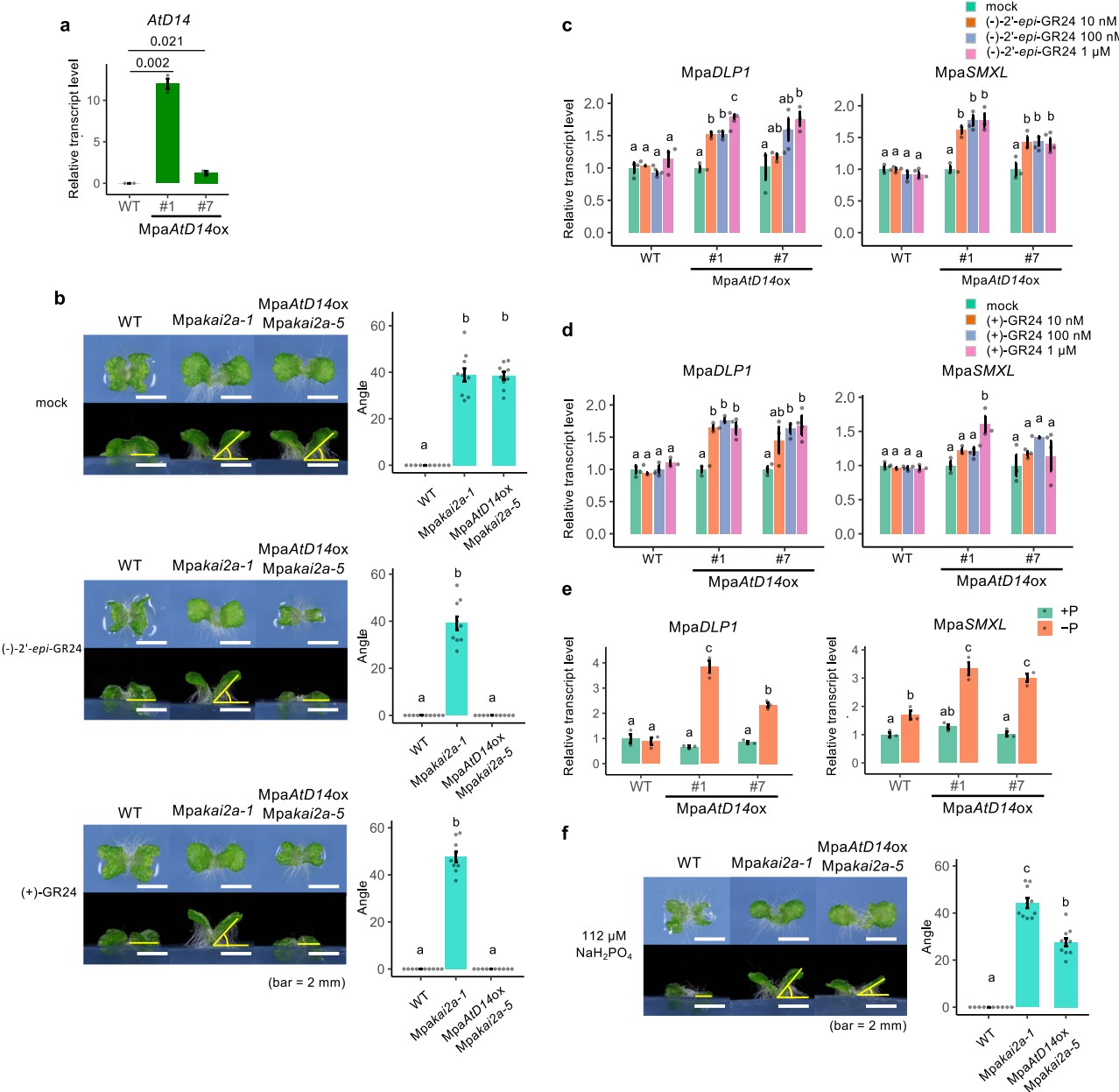

**Fig. 7 Heterologous expression of *AtD14*, the SL receptor of Arabidopsis, renders *M. paleacea* cells BSB-responsive. a** Transcript level of *AtD14* introduced into *M. paleacea*. Data are means ± SD ($n = 3$ biologically independent samples), and $p$ values in Welch's $t$-test, two tailed are indicated. **b** Complementation of Mpa*kai2a* phenotypes in Mpa*AtD14*ox/Mpa*kai2a* (*M. paleacea*) lines by addition of 1 μM (−)-2′-*epi*-GR24 or 1 μM (+)-GR24. Data are means ± SD ($n = 10$ biologically independent plants). **c, d** Induction of Mpa*DLP1* and Mpa*SMXL*, marker genes of KAI2-dependent signaling, in response to SL treatment. Transcript level of Mpa*DLP1* and Mpa*SMXL* in WT and Mpa*AtD14*ox lines in *M. paleacea* after (−)-2′-*epi*-GR24 (**c**) or (+)-GR24 (**d**) treatment are shown. Data in **c, d** are means ± SD ($n = 3$ biologically independent samples). **e** Induction of Mpa*DLP1* and Mpa*SMXL* transcript level by phosphate starvation in Mpa*AtD14*ox lines of *M. paleacea*. Data are means ± SD ($n = 3$ biologically independent samples). **f** Complementation of Mpa*kai2a* phenotypes in Mpa*AtD14*ox/Mpa*kai2a* (*M. paleacea*) lines grown on media with a reduced phosphorus level. Data are means ± SD ($n = 10$ biologically independent plants). The HSD test was used for multiple comparisons in **b**–**f**. Statistical differences ($p$ values < 0.05) are indicated by different letters in **b**–**f**.

cognate receptor is present. This strongly supports the hypothesis that the evolution of SLs as an endogenous hormone is only dependent on the evolution of a cognate receptor in the KAI2 clade.

## Discussion

In flowering plants, SLs play dual roles as a class of phytohormones and as rhizosphere signaling chemicals. As phytohormones, SLs control various aspects of growth and

development while as rhizosphere signaling molecules, SLs control symbiosis with AM fungi and parasitism of root parasitic plants[6–8,50]. AM symbiosis is found in most extant land plants, which comprise two main monophyletic clades, the non-vascular bryophytes and the vascular tracheophytes and was present in the last common ancestor of land plants more than 400 million years ago[3] In this study, we showed that the SL, BSB, is produced in a wide range of plant species, including flowering plants, indicating that BSB is an ancestral and conserved form of SLs. In *M.*

*paleacea*, BSB acts as a rhizosphere signaling molecule essential for symbiosis with AM fungi. Thus, it can be concluded that the function of SLs as signaling molecules to communicate with AM fungi was already established in the common ancestor of the bryophytes and vascular plants.

AM symbiosis was an innovation by the first land plants to acquire minerals, in particular phosphate, from the nutrient-poor soil they faced during the colonization of emerged land. In all mutualistic systems, control mechanisms have evolved on both the host and symbiont sides to optimize the benefits of the symbiosis[51]. In flowering plants, SL synthesis is regulated by phosphate availability[32,52,53]. Upon phosphate starvation, SL biosynthesis and transporter genes are enhanced resulting in an increase in the level of SLs exuded to the rhizosphere[34]. This leads to enhanced AM symbiosis and consequently, to increased phosphate supply. We showed that phosphate-dependent control of SL synthesis occurs in the liverwort *M. paleacea* and the hornworts *Anthoceros agrestis*. These results imply that the control of AM symbiosis through phosphate-dependent SL synthesis was already present in the most common ancestors of bryophytes and vascular plants.

The *CCD8* and *MAX1* have been lost from the *M. polymorpha* genome, whereas *KAI2A*, *KAI2B*, and *MAX2* are conserved in *M. polymorpha*. The developmental phenotypes observed in *M. paleacea kai2a* and *max2* mutants resemble the ones observed in *M. polymorpha*. Although we cannot rule out that BSB does play an endogenous role in *M. paleacea*, the absence of observed growth phenotypes in the Mpa*max1* and Mpa*ccd8a/8b* mutant of *M. paleacea*, and the loss of both *CCD8* and *MAX1* in *M. polymorpha*, a non-symbiotic species, suggest that its function is exclusively symbiotic in liverworts. Even more certain is the disconnection between BSB and the MAX2-dependent signaling pathway. Loss of *CCD8* is also observed in the water ferns *Azolla filiculoides* and *Salvinia cucullata* that have also lost AM symbiosis[54]. Such a correlation between gene and trait losses is indicative of a specific function of the genes, here in AM symbiosis. Thus, it can be inferred that the metabolites from the CCD8/MAX1 SL biosynthesis pathway and MAX2-dependent signaling pathway were originally independent. The role of the CCD8-derived metabolites as endogenous plant hormones evolved during seed plant diversification by connecting them to receptors from the KAI2/D14 family. Extensive phylogenetic analysis revealed that D14 evolved through putative neo-functionalization following gene duplications of the KAI2 family members in the common ancestor of seed plants, leading to its function as cognate receptor for SLs as plant hormones regulating multiple developmental traits in flowering plants[13,43,50]. In this study, we mimicked the evolution of such a hormonal system by introducing the *D14* from Arabidopsis into *M. paleacea* and showed that endogenous BSB and exogenously supplied SL analogs are perceived by the introduced D14 and trigger signal transduction in these engineered lines through MAX2-dependent signaling.

Despite the lack of *MAX1*, molecules with an SL function are synthesized through CCD8 in *P. patens*[16]. CCD8-derived metabolites have also been recruited for quorum sensing-like functions in *P. patens*[15]. These metabolites are currently elusive but the absence of *MAX1* in *P. patens* suggests that those are neither CLA nor BSB. Interestingly, the *KAI2* genes are highly duplicated in *P. patens* and some of them are involved in the perception of the unknown CCD8-derived metabolites, but their signaling is MAX2-independent[55,56]. These findings imply that *P. patens* generated both the metabolic pathway and the signaling system independently from other bryophytes after the loss of the *MAX1* gene. Similar acquisition of signaling systems occurred in root parasitic plants, in which the *KAI2* genes duplicated massively[50].

Some of the *KAI2* paralogs were identified as extremely sensitive SL receptors that recognize SLs from host plants[50,57,58]. Therefore, the generation of SL receptors from KAI2 and of molecules with an SL function occurred repeatedly and independently during the evolution of land plants.

The origin of the complex hormonal networks observed in extant flowering plants has been extensively studied by combining reverse genetics conducted in bryophytes and phylogenetic studies on the entire green lineage[59,60]. These networks seem to be highly conserved and thus under strong purifying selection in angiosperms where they have been first discovered and dissected. Each of these networks can be divided in three layers: ligand, receptors and signaling pathways. From our work and previous studies, we propose a general model for their evolution.

First, each layer of the network may evolve independently from each other in an unconstrained way, leading to lineages with both a biosynthetic pathway and a functional signaling pathway present. From this state, we propose that the evolution of a receptor is sufficient to evolve a fully functional hormonal network. Although land plants possess a specific auxin biosynthesis pathway, auxin production is widespread among eukaryotes, including chlorophyte and streptophyte algae. Streptophyte algae also possess the signaling network but lack the TIR1-type receptors which evolved in land plants[61]. ABA biosynthesis has been observed in chlorophyte algae[61–63] while the ortholog of the ABA receptor evolved in the common ancestor of land plants and the Zygnematophyceae, which are the closest algal clade relative to land plants, with an ancestral function presumed to be ABA independent[64,65]. The ABA-receptor ortholog in land plants acquired the ability to bind ABA which was sufficient to recruit its downstream signaling pathway[26,66]. Here, we proposed and experimentally demonstrated that the gain of a receptor was sufficient to recruit SL as hormonal signal. Why such connections between the product of an existing biosynthetic pathway and a signaling pathway emerge and are selected remains an open question. An attractive view is that SLs being exuded under phosphate deprivation conditions may have been first recruited as a rhizospheric signal in plant communities and later as an endogenous mobile signal. Stepwise recruitment of non-connected biosynthetic and signaling pathways into integrated processes might represent a general mechanism for the evolution of intercellular and inter-organism communication.

## Methods

**Plant materials and growth conditions**. *Marchantia paleacea* ssp. *diptera*, *M. polymorpha*, *M. pinnata*, *M. emarginata* and *Phaeoceros carolinianus* were harvested in the wild in Japan (*Marchantia paleacea* ssp. *diptera* was collected in Nara prefecture, the others were collected in Hiroshima prefecture). A *Physcomitrium patens* line previously reported was used[67]. *Dryopteris erythrosora* and *Matteuccia struthiopteris* plants were purchased from a garden store. Seeds of *Asparagus officinalis* (cv. Shower, Takii, Japan) and *Arachis hypogaea* (cv. Chiba-handachi, Watanabe Noji, Japan) were used. *M. polymorpha* and *M. paleacea* were grown at 22 °C under continuous light on 1/2 Gamborg's B5 medium (pH 5.5) with 1% agar (Guaranteed Reagent; FUJIFILM WAKO Chemicals, Japan) or on expanded vermiculite fertilized with 1/2000 hyponex (HYPONEX Japan) every 2 weeks. *P. patens* plants were grown at 25 °C under continuous light on BCDAT soiled medium[68]. *A. agrestis* (Bonn strain) was grown on Half-strength Knop-II media (125 mg/l KNO$_3$, 500 mg/l Ca(NO$_3$)$_2$ 4H$_2$O, 125 mg/l Mg(SO$_4$)$_2$ 7H$_2$O, 125 mg/l KH$_2$PO$_4$, and 10 mg/l FeCl$_3$ 6H$_2$O; pH5.5) containing 1% agar (FUJIFILM Wako Pure Chemical Corporation, Japan).

For phosphate-deficient treatment (Fig. 4a), Gemmae of *M. paleacea* or thallus of *A. agrestis* were pre-incubated on half-strength Gamborg's B5 medium with 1.0% agar for 2 weeks at 22 °C under continuous light. After 2 weeks, the thallus was transferred to the medium with or without phosphate and incubated for another week. Resultant plants were harvested and used for expression analysis.

For low phosphate treatment (Figs. 6c, d and 7f), Gemmae of *M. paleacea* were incubated on half-strength Gamborg's B5 medium with 1.0% agar with reduced phosphorus concentration (5-fold or 10-fold low phosphate concentration) for the appropriate time as described in each legend.

To analyze thallus phenotypes, gemmae were grown on half-strength Gamborg's B5 medium with 1.0% agar or the medium with a reduced phosphorus concentration, at 22 °C under continuous light for 21 days. Resultant plants were used for the analysis of thallus phenotypes. To accurately measure the area and rhizoid length, plants were flattened with a coverslip on a glass slide, and images were taken. To measure the rhizoid length, the average lengths of the three longest rhizoids of each thallus were used as a representative rhizoid length for each thallus. To measure the angle between the thallus and the growth medium, images were taken from the side. For all measurements, the images were analyzed using ImageJ[46].

**Plasmid construction and plant transformation.** To construct vectors to generate CRISPR-Cas9 mutants, oligo DNAs containing the target sequences and the complementary sequences were annealed and introduced into pMpGE_En01 (*ccd8a, ccd8b, kai2a, kai2b*) or pMpGE_En03 (*max1, max2*). Subsequently, they were inserted into pMpGE010 (*ccd8b-1, -2, kai2a-2, kai2b-1, -2*) or pMpGE011 (*ccd8a-1, -3, max1-1, -2, kai2a-1, max2-1, -2*) by Gateway LR reactions.

Transformation of *M. polymorpha* was performed as described[69], and slight modifications were applied for the transformation of *M. paleacea*. For transformation of *M. polymorpha*, 10- to 14-day-old thalli grown from gemmae were transferred to fresh media and incubated for 2–3 days before *Agrobacterium* infection. For transformation of *M. paleacea*, 1- to 2-month-old thalli were cut and incubated on fresh media for 5–7 days before infection. To generate the Mpa*kai2akai2b* double mutant, *KAI2A* guide RNA inserted in pMpGE011 was introduced to *kai2b-1*. To generate the Mpa*ccd8accd8b* double mutant, *CCD8A* guide RNA inserted in pMpGE011 and *CCD8B* guide RNA inserted in pMpGE010 were introduced to WT thalli at the same time (Mpa*ccd8a-2/8b-3*) or *CCD8B* guide RNA inserted in pMpGE010 was introduced into *ccd8a-1* (Mpa*ccd8a-1/8b-4*).

**Analysis of CCD8A in *Marchantiopsida* species.** Genomic DNA of *Marchantia chenopoda*, *M. emarginata* and *M. pinnata* was extracted as follow: 1 cm² of thalli was ground in 400 μl of extraction buffer (200 mM Tris-HCl, 25 mM EDTA, 250 mM NaCl, 0.5% SDS) and centrifuged for 5 min. In total, 300 μl of isopropanol was added to the supernatant and centrifuged 5 min. Supernatant was discarded and pellet dried for 15 min. DNA was resuspended in 100 μl of water. In total, 2.5 kb of the *CCD8A* gene was PCR-amplified using GoTaq polymerase (Promega) according to the manufacturer's protocol and the primers MpCCD8-F and MpCCD8-R (Supplementary Table 4). Purified PCR products were Sanger-sequenced at Eurofins (Germany) using the same primers.

**SL analysis in exudates and plants.** *M. paleacea*, *M. polymorpha*, *M. pinnata*, *M. emarginata* and *P. carolinianus* were grown in 10 cm diameter pots filled with vermiculite under a 15 h light (50 μmol m⁻² s⁻¹)/9 h dark photoperiod at 23 °C in a growth room. *D. erythrosora* and *M. struthiopteris* plants were grown in a glasshouse with temperature control and under a natural photoperiod, at 23 °C in the day and 18 °C at night during spring to summer. *A. officinalis* and *A. hypogaea* seedlings were grown at 25 °C in the day and 20 °C at night. These plants were grown using 1/2000 Hyponex and tap water. At least 2 weeks after growing in tap water, 200 ml per pot of water was poured on the plants (*Marchantia* and *Phaeoceros*) or the surface of the vermiculite (other plants), and collected from the bottom of the pot. The exudates were extracted with ethyl acetate. For SL quantification, 5 ng each of [1-¹³CH₃]*rac*-CLA and [3a,4,4,5,5,6'-D₆]4DO were added as internal standards prior to the extraction. Endogenous SLs in *M. paleacea* and *P. patens* plants were extracted with acetone. SLs were analyzed by LC-MS/MS using a triple quadrupole/linear ion trap instrument (QTRAP 5500; AB Sciex, USA)[23]. HPLC separation was performed on a UHPLC (Nexera X2; Shimadzu, Japan) equipped with an ODS column (Kinetex C18, φ 2.1 × 150 mm, 1.7 μm; Phenomenex, USA). The column oven temperature was maintained at 30 °C. The mobile phase consisted of acetonitrile and water, both of which contained 0.1% (v/v) acetic acid. HPLC separation was conducted with a linear gradient of 35% acetonitrile (0 min) to 95% acetonitrile (20 min) at flow rate of 0.2 ml min⁻¹. *rac*-CL, *rac*-CLA, [1-¹³CH₃]*rac*-CLA and [3a,4,4,5,5,6'-D₆]4DO were synthesized in previous study[70].

**Seed germination assay.** *Orobanche minor* seeds were collected from mature plants that parasitized red clover (*Trifolium pratense* L.) grown in Tochigi Prefecture, Japan in June 2017. *Phelipanche ramosa* seeds (Genetic group 2a) collected from mature plants parasitizing hemp (*Cannabis sativa*) were supplied by Prof. Philippe Delavault (University of Nantes, France). *Striga hermonthica* seeds were supplied by Prof. A. G. T. Babiker (Sudan University of Science and Technology, Sudan). Seeds of *O. minor*, *P. ramosa* and *S. hermonthica* were surface-sterilized in 1% NaOCl containing 0.1% Tween-20 for 5 min. The seeds were then rinsed ten times with sterile distilled water and air-dried. Twenty to 40 seeds were sown on a 5 mm-glass fiber disk prepared by a hole puncher (Whatman GF/A, UK). Twenty disks were placed in a 5 cm-Petri dish lined with two layers of filter paper wetted with 1.5 ml of distilled water. The Petri dishes were sealed with parafilm, enclosed in polyethylene bags and incubated in the dark at 23 °C for 5 days (*O. minor*), 18 °C for 7 days (*P. ramosa*) or 30 °C for 15 days (*S. hermonthica*). After conditioning, the glass fiber disks were blotted to remove excess water. Each group of three disks

was transferred to a separate 5 cm-Petri dish lined with two layers of filter paper and wetted with 0.6 ml of test solutions. The Petri dishes were sealed, enclosed in polyethylene bags, and placed in the dark at 23 °C for 5 days (*O. minor*), 20 °C for 6 days (*P. ramosa*) or 30 °C for 3 days (*S. hermonthica*). The germination rate was determined under a ×20 binocular dissecting microscope. For germination assay shown in Fig. 4d, *M. paleacea* gemmae were grown on half-strength Gamborg's B5 medium with 1.0% agar at 22 °C under continuous light for 3 weeks, then on the liquid medium with or without phosphate for another 2 weeks. The liquid media were directly used for germination assay of *Orobanche minor* seeds.

**Isolation and purification of bryosymbiol (BSB).** *M. paleacea* was grown on vermiculite in a plastic container (with holes in the bottom, 38 × 25 × 14 cm, W × L × H) using tap water. Eighteen containers were placed under a 14 h light (60 μmol m⁻² s⁻¹)/10 h dark photoperiod at 24 °C in a growth room. Water was poured onto the plants and circulated every 10 min per hour using an aquarium pump. The exudates released into the water were adsorbed on the activated charcoal (2 g per 3 l) in the pump. Fresh activated charcoal was replaced every 2 days. The collected charcoal was eluted with acetone. After evaporation of the acetone in vacuo, the aqueous residue was extracted with EtOAc. The EtOAc phase was washed with 0.2 M K₂HPO₄ (pH 8.3), dried over anhydrous MgSO₄, and concentrated in vacuo. The concentrated samples were kept at 4 °C until use. The crude EtOAc extract (107.1 mg), collected over 5 weeks, was subjected to silica gel column chromatography (55 g) with stepwise elution of *n*-hexane–EtOAc (100:0 to 0:100, v/v, 10% step) to give fractions 1–11. Fractions 6 (50% EtOAc, 13.2 mg) containing a novel SL was subjected to silica gel column chromatography (20 g) using *n*-hexane–EtOAc (60:40, v/v). Fractions were collected every 10 ml. Fractions 9–18 were found to contain a novel SL based on LC-MS/MS and GC-MS analyses. Fractions 9–18 were combined (3.17 mg) and purified by HPLC on an ODS column (Kinetex C18, 4.6 × 250 mm, 5 μm; Phenomenex, USA) with a MeCN/H₂O gradient system (30:70 to 100:00 over 40 min) at a flow rate of 1 ml/min. The column temperature was set to 30 °C. The fraction eluted as a single peak at 28.6 min (detection at 248 nm) was collected. This fraction was further purified by HPLC on a Develosil ODS-CN column (4.6 × 250 mm, 5 μm; Nomura Chemicals, Japan) with isocratic 70% MeCN/H₂O at a flow rate of 0.8 ml min⁻¹ to afford BSB (0.33 mg, *Rt* 21.1 min, detection at 248 nm).

**Structural determination of BSB.** BSB was obtained as a white amorphous solid with a molecular formula of C₁₉H₂₄O₆ established by the HR-ESI–TOF–MS peak at *m/z* 349.1657 ([M + H]⁺, calculated for C₁₉H₂₅O₆, *m/z*: 349.1651) (Xevo G2-XS QTof, Waters, USA), with eight degrees of unsaturation (Supplementary Fig. 3). EI-GC/MS (JMS-Q1000GC/K9, JEOL, Japan; DB-5 column, Agilent, USA) gave a molecular ion peak at *m/z* 348 [M]⁺, with fragment ion peaks at *m/z* 234, 219, 153, 123, and 97 (the D-ring fragment) (Supplementary Fig. 4). The ¹H and ¹³C NMR spectra (JMN-ECA-500 spectrometer, JEOL; in CDCl₃ (δ_H 7.26, δ_C 77.0)) aided with DEPT and HMQC experiments revealed 19 carbon signals including two α,β-unsaturated ester carbonyls, two trisubstituted olefins, one tetrasubstituted olefin, three sp³ oxymethines, three sp³ methylenes, two methyls, one quaternary sp³ carbon, and two allylic methyls (Supplementary Table 2). The ¹H-¹H COSY and HMBC correlations from H-2′, H-3′, H-6′, and H-7′ indicated the presence of the conserved enol ether-bridged methylbutenolide (the D-ring) (Supplementary Fig. 5 and Supplementary Table 2). The HMBC correlations of H-6′ to C-2, C-3, and C-4 and of H-5 to C-4 established the presence of α-methylene-β-hydroxy-γ-butyr-olactone ring (the C-ring). The ¹H-¹H COSY and HMBC correlations from three methyls at H-12, H-13, and H-14 and from three sp³ methylenes at H-8, H-9, and H-10 showed a 2,6,6-trimethylcyclohex-1-en-1-yl moiety (the A-ring). The two substructures, the A- and the CD-parts, were connected together at C-5 and C-6 as indicated by the HMBC correlation of H-5 to C-6. Therefore, the planar chemical structure of BSB was determined as (*E*)-5-((4-hydroxy-2-oxo-5-(2,6,6-tri-methylcyclohex-1-en-1-yl)dihydrofuran-3(2*H*)-ylidene)methoxy)-3-methylfuran-2(5*H*)-one. The NOESY correlations between H-12/H-13 methyls and H-5 oxy-methine, and between H-14 methyl and H-4 oxymethine suggested the relative stereochemistry of H-4 and H-5 is anti-orientation. DFT-GIAO NMR calculations of the possible four isomers with respect to C-4, C-5, and C-2′ were carried out at the PCM/mPW1PW91/6-31 + G**//B3LYP/6-31G* level of theory, in which the 4S*,5R*,2′R* isomer was identified as the correct one by DP4+ analysis with 95.86% probability (Supplementary Fig. 6). The CD spectrum of natural BSB (J-820W spectropolarimeter, JASCO, Japan; in acetonitrile (*c* 0.002)) had a positive and negative Cotton effect around 224 nm (Δε 27.4) and 266 nm (Δε −4.44), respectively, indicating that BSB has a 2′*R* configuration as previously identified natural SLs (Supplementary Fig. 7). TDDFT-ECD calculations for 4S, 5R, 2′R isomer and its enantiomer were run with several combinations of functionals (ωB97X-D, B3LYP, BH&HLYP,) and the TZVP basis set using IEFPCM solvent model for acetonitrile (Supplementary Fig. 8). The all calculated spectra for the 4S, 5R, 2′R isomer showed a small negative and a large positive Cotton effect respectively between 250 and 270 nm and between 220 and 230 nm as observed in the ECD spectrum of natural BSB. Thus, the absolute configuration of BSB was assigned as 4S, 5R, 2′R. In an attempt to further confirm the stereochemistry by X-ray crystallography, this compound was decomposed due to its instability during recrystallization. Structure confirmation by chemical synthesis also failed due to our inability to introduce a formyl group at the alpha-carbon in the butyrolactone

ring. In fact, no reports were found on the α-acylation of β-hydroxybutyrolactone in literature.

**Expression of MpaMAX1 in yeast.** Total RNAs were extracted from whole *M. paleacea* plants with rhizoids using the RNeasy Plant Mini Kit (Qiagen, Germany) and used to synthesize single-stranded cDNAs with the SuperScript III First-Strand Synthesis System (Invitrogen, USA). PCR amplification was performed with PrimeSTAR HS DNA polymerase (TaKaRa Bio, Japan) using the primers Mpa-MAX1_cDNA_F and MpaMAX1_cDNA_R (Supplementary Table 4). The MpaMAX1 cDNA was cloned into the *BamH I* and *Kpn I* sites of the pYeDP60 vector with the GeneArt Seamless Cloning and Assembly Enzyme Mix (Invitrogen, USA) using the primers MpaMAX1_cloning_F and MpaMAX1_cloning_R (Supplementary Table 4). At least four clones were sequenced to check for errors in the PCR. The accession number of MpaMAX1 was deposited as LC553047. Heterologous expression of MpaMAX1 was carried out using the WAT11 strain of yeast (*Saccharomyces cerevisiae*)[21,69]. Transformed yeasts were incubated in SGI medium at 30 °C until plateau, then inoculated into SLI medium and grown at 28 °C until the cell density reached $5 \times 10^7$ cells ml$^{-1}$. To prepare microsomal proteins, collected yeast cells were broken using a high-pressure homogenizer (EmulsiFlex-B15; Avestin, Inc., Canada). The broken cells were centrifuged at $10,000 \times g$ for 15 min and the resulting supernatant was centrifuged at $100,000 \times g$ for 1 h (himac CP80NX; HITACHI, Japan). Microsomes (100 μl) were incubated with *rac*-CL (0.25 to 33 μM) and NADPH (500 μM) at 28 °C for 10 min. The reaction mixtures were extracted with ethyl acetate, dried over anhydrous Na$_2$SO$_4$ and then subjected to LC-MS/MS analysis. The product BSB was quantified with the peak area in the transitions of *m/z* 349 to 97 by assuming the same ion intensity as that of 4DO. The kinetics parameter was determined using triplicate samples and calculated by the Michaelis-Menten equation using SigmaPlot 14 (Systat Software, USA). The $V_{max}$ of BSB from *rac*-CL was $92.8 \pm 8.4$ nM min$^{-1}$ and the Michaelis constant was $4.78 \pm 1.22$ μM.

**Mycorrhization assays.** Thalli of *Marchantia paleacea*, *Marchantia emarginata*, *Marchantia pinnata* and *Marchantia chenopoda* were grown on a substrate composed of 50% zeolite and 50% sand (0.7–1.3 mm) in 7 × 7 × 8 cm pots (five thalli per pot). Each pot was inoculated with *ca*. 1000 sterile spores of *Rhizophagus irregularis* DAOM 197198 (Agronutrition, Labège, France) and grown with a 16 h/8 h photoperiod at 22 °C/20 °C. Pots were watered once per week with half-strength Long Ashton medium containing 7.5 μM of phosphate[32]. For SL complementation assays, 10 ml of half-strength Long Ashton solution with 10$^{-7}$ M GR24 or 0.1% acetone (control) was added to each pot weekly, starting on the day of the inoculation. Hyphal branching activity of BSB on *Gigaspora margarita* was evaluated using the paper disk diffusion method[35]. Spores of *G. margarita* MAFF520054 were surface-sterilized with 0.2% NaClO and 0.05% Triton X-100 for 8 min. The spores were then rinsed eight times with sterile MilliQ water. One or two spores were inserted into a 0.2% Phytagel gel (Sigma-Aldrich, USA) containing 3 mM MgSO$_4$ in 60-mm plastic Petri dishes. The dishes were incubated vertically for 6 days in a 2% CO$_2$ incubator at 32 °C. Test samples were first dissolved in acetone and then diluted with 70% ethanol in water. Two paper disks (6 mm in diameter; ADVANTEC, Japan) loaded with test sample solutions were placed in front of the tips of the secondary hyphae. The control was 70% ethanol in water. The hyphal branching patterns were observed under a stereoscopic microscope at 24 h after treatment. The sample was scored as positive if the clusters of hyphal branches of higher order are induced from the treated secondary hyphae located proximal to the paper disks. No hyphae or an occasional single branching hypha is induced in the control. Assay was repeated three times, using three dishes for each concentration. The activity was evaluated by determining the minimum effective concentration using serial 10-fold dilutions of the test samples.

To establish the mycorrhizal status of the different Marchantia species, thalli (8 weeks post-inoculation) were embedded in 6% agarose and 0.1 mm transversal sections were prepared using a Leica vt1000s vibratome. Sections were incubated overnight in PBS containing 1 μg/ml WGA-FITC (Sigma) and pictures were taken with a Zeiss Axiozoom V16 microscope.

For the mycorrhizal phenotyping of the mutants, thalli were harvested 6 weeks post-inoculation and longitudinal sections were cleared in 10% KOH overnight and ink stained for 30 min in 5% Sheaffer ink, 5% acetic acid. Sections were observed under a microscope and infection points were scored for each thallus.

For the quantification of the symbiosis under different phosphate regimes, plants were watered weekly with Long Ashton solutions containing 0.015, 0.5, 1 or 2.5 mM of NaH$_2$PO$_4$ and harvested at 5 or 8 weeks post-inoculation. Thalli were cleared of chlorophyll using ethanol and distance between notches and colonization zone (naturally colored with a purple pigment) were scored on scanner pictures.

All experiments were repeated three times with similar results.

**Phylogenetic analysis.** Homologs of the reference genes *CCD8* and *MAX1* of *Arabidopsis thaliana* or *Medicago truncatula* were retrieved in a custom database composed of 197 species covering the main land plant lineages (Supplementary Data 1) and including, for Lycophytes, Monilophytes and Bryophytes, predicted

coding sequences of transcriptomic data from the 1KP project[16,57]. Searches were perform using the tblastn v2.9.0+ algorithm with default parameters and an e-value threshold of 1$^{e-10}$.

For each gene, their putative homolog coding sequences were aligned using MAFFT v7.313 with default parameters and the resulting alignments trimmed to remove positions with more than 80% gaps using trimAl v1.4.1. Then, alignments were subjected to maximum likelihood (ML) analysis using IQ-TREE v1.6.7 and branch support tested with 10,000 replicates of SH-aLRT and UltraFast Bootstraps. Prior to ML analysis, the best-fitting evolutionary model was tested using ModelFinder and retained according to the Bayesian Information Criteria. Trees were visualized and annotated using the iTOL v5.6.3[71].

**RNA extraction and expression analysis.** RNA was extracted from frozen plant samples using the NucleoSpin RNA Plant kit (MACHEREY-NAGEL). cDNAs were synthesized using SuperScript VILO (Invitrogen). Quantitative PCR was performed using the KOD SYBR® qPCR Mix (Toyobo Life Science) with Light Cycler 480 II (Roche). Primers used to amplify the cDNAs are shown in Supplementary Table 6. The ACTIN gene of each species was used as a standard.

**RNA-seq analysis.** Gemmae of WT, Mpa*ccb8a-1/b-4*, and Mpa*ccb8a-2/b-3* were incubated on half-strength Gamborg's B5 medium for 2 weeks at 22 °C under continuous light. After 2 week-culture, the thalli were transferred to the medium with or without phosphate. Three biological replicates were prepared for each genotype. Total RNA was extracted using the NucleoSpin RNA Plant kit (Macherey-Nagel, Germany). mRNA was isolated from the total RNA with an NEBNext poly(A) mRNA Magnetic Isolation Module Kit (New England Biolabs, MA, USA). RNA-seq libraries were synthesized using an NEBNext Ultra RNA Library Prep Kit for Illumina and an NEBNext Adaptor for Illumina Kit (New England Biolabs, MA, USA). Paired-end sequencing was performed using the HiSeqXten (Illumina, CA, USA).

The reads were mapped onto the *Marchantia paleacea* genome[3] assembly using HISAT2[72] for Galaxy (https://usegalaxy.org) with default parameters. Read counts were calculated by Feature Counts for Galaxy. To identify differentially expressed genes, *q* values were calculated by the TCC R package[73], and genes with *q* value <0.05 and log2-fold change >1 or log2-fold change <−1 were selected as up-or down-regulated genes. Summary statistics of RNA-seq analysis are available in Supplementary Data 3.

**Differential scanning fluorimetry (DSF) analysis.** The coding sequences for MpaKAI2A and MpaKAI2B were amplified from cDNA synthesized from total mRNA of the *Marchantia paleacea* thalli using the following primer sets, Mpa-KAI2A_cDNA_F and MpaKAI2A_cDNA_R for MpaKAI2A and Mpa-KAI2B_cDNA_F and MpaKAI2B_cDNA_R for MpaKAI2B (Supplementary Table 4). The PCR products were cloned into the modified pET28 vector using the In-Fusion HD Cloning Kit (Clontech) to generate pET28M-MpaKAI2A and pET28M-MpaKAI2B. *Escherichia coli* strain BL21 (DE3) (Takara) was used for recombinant protein expression. Overnight cultures (3 ml) in LB liquid medium containing 100 μg/ml kanamycin were inoculated into fresh LB medium (1 l) containing 100 μg/ml kanamycin and incubated at 37 °C until the OD600 value reached to 0.6–0.8. Then, IPTG was added to 200 μM, and the cells were further incubated at 25 °C overnight. The cells were collected by centrifugation at 4000 rpm for 15 min. The pellets were resuspended in 30 ml of buffer A (20 mM Tris-HCl (pH 8.0) and 200 mM NaCl). The cells were crushed using a cell disruption device (UD-200, TOMY SEIKO). The soluble fraction was loaded on Ni Sepharose 6B fast flow (Cytiva) pre-equilibrated with buffer A. The bound Mpa-KAI2A and MpaKAI2B proteins were eluted with 5 ml of buffer A containing 125 mM of imidazole. The purified proteins were concentrated using an Amicon Ultra-410K (Millipore).

For DSF analysis, 20 μl reaction mixtures containing 10 μg protein, 0.015 μl Sypro Orange and GR24s with final concentrations of 250, 100 or 50 μM, were prepared in 96-well plates. The final acetone concentration in the reaction mixture was 5%. 1xPBS buffer (pH 7.4) containing 5% acetone was used in the control reaction. DSF experiments were performed using a Light Cycler 480 II (Roche). Sypro Orange (Invitrogen) was used as the reporter dye. Samples were incubated at 25 °C for 10 min, then, the fluorescence wavelength was detected continuously from 30 to 95 °C. The denaturation curve was obtained using MxPro software (Agilent).

**Generation of AtD14-overexpressing plants.** To generate *AtD14*ox lines, the AtD14 coding sequence cloned into pENTR/D-TOPO vector[42] was inserted into pMpGWB103 or pMpGWB303 by Gateway LR reactions. The MpEFpro: AtD14 vector was introduced into regenerating thalli of Tak-1 or WT of *M. paleacea* as described above.

To generate Mp*AtD14*ox/Mp*kai2a* or Mpa*AtD14*ox/Mpa*kai2a* line, Mp*KAI2A* or Mpa*KAI2A* was knocked out in Mp*AtD14*ox or Mpa*AtD14*ox lines. Mp*KAI2A* guide RNA inserted in pMpGE010[46] or Mpa*KAI2A* guide RNA inserted in pMpGE011 were introduced into Mp*AtD14*ox or Mpa*AtD14*ox lines, respectively.

**Application of GR24**. Synthetic SL analog *rac*-GR24 was obtained from Chiralix B.V. (Netherland). The enantiomers of GR24 were obtained by optical resolution from *rac*-GR24 stationary phase (Chiralpak IC-3, 250 × 4.6 mm, Daicel, Japan) under normal-phase using hexane/isopropanol (70/30) by HPLC. (−)-2′-*epi*-GR24 and (+)-GR24 were identified by CD analysis. *rac*-GR24, (−)-2′-*epi*-GR24 and (+)-GR24 dissolved in acetone were added to half-strength Gamborg's B5 medium with 1.0 % agar. Thalli grown from gemmae for 2 weeks were incubated on the *rac*-GR24, (−)-2′-*epi*-GR24, (+)-GR24, or mock containing media for 6 h.

**Statistical analysis**. We used Welch's *t*-test or Tukey's honestly significant difference test to evaluate statistical significance. Experimental sample sizes and statistical methods detail are given in the each legend.

**Reporting summary**. Further information on research design is available in the Nature Research Reporting Summary linked to this article.

## Data availability

The accession numbers of genes analyzed in this study are shown in Supplementary Table 5. Transcriptome data were deposited in the DDBJ and are available through the Sequence Read Archive (SRA) with the identifier DRA012476. LC-MS/MS data were deposited in Metabolights with the identifier MTBLS5064. The authors declare that all data supporting the findings of this study are available within the manuscript and its Supplementary files or are from the corresponding author upon request. Source data are provided with this paper.

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

## Acknowledgements

We thank Philippe Delavault (University of Nantes) and A.G.T. Babiker (Sudan University of Science and Technology) for *Phelipanche ramosa* and *Striga hermonthica* seeds, respectively. We are grateful to the genotoul bioinformatics platform Toulouse Occitanie (Bioinfo Genotoul, doi: 10.15454/1.5572369328961167E12) for providing computing resources. Researches conducted by J.Kyozuka lab and T.Nomura lab were supported by a Grants-in-Aid from the Ministry of Education, Culture, Sports, Science, and Technology, Japan (20H05684 and 17H06475 to J.Kyozuka, 19K05838 to T.Nomura, 18K05452 to X.X.) and Canon Foundation to J.Kyozuka. Research conducted in P.-M.D. lab was supported by the Agence Nationale de la Recherche (ANR) grant EVOLSYM (ANR-17-CE20-0006-01) and was supported by the project Engineering Nitrogen Symbiosis for Africa (ENSA) currently supported through a grant to the University of Cambridge by the Bill & Melinda Gates Foundation (OPP1172165) and the UK Government's Department for International Development (DFID). The Laboratoire de Recherche en Sciences Végétales (LRSV) belongs to the TULIP Laboratoire d'Excellence (ANR-10-LABX-41).

## Author contributions

J.Kyozuka, P.-M.D., M.K.R. and T.Nomura designed the research and wrote the article. K.K., M.K.R., S.S., Y.M. and A.K. produced mutants and analyzed phenotypes. K.K., S.S., Y.M., A.K., Y.L., H.S. and H.K. examined gene expression. S.S., M.K.R., T.Nakagawa and P.-M.D. analyzed A.M. symbiosis. C.L. and J.Keller conducted phylogenetic analysis. S.S., A.Y., K.Y., K.M., S.Y. and Y.T. analyzed BSB and enzyme activity. X.X., K.U. and K.A. determined the chemical structure of BSB. K.S., M.S., T.Nakagawa and T.Nishiyama prepared materials.

## Competing interests

The authors declare no competing interests.
