## [Peer Review File · Nature Communications]

An Ancestral Function of Strigolactones as Symbiotic Rhizosphere SignalsREVIEWER COMMENTS

Reviewer #1 (Remarks to the Author):

Kodama et al

An Ancestral Function of Strigolactones as Symbiotic Rhizosphere Signals

This manuscript describes the identification of a small secreted compound, BSB, whose function in *Marchantia paleacea* appears to be in a communication pathway with colonizing arbuscular mycorrhizal fungi. Its production is increased by phosphate deficiency consistent with the proposed role. As this compound is produced both in liverworts and seed plants, it appears it was produced in the ancestral land plant. The incorporation of similar molecules, i.e. strigolactones, into a signalling pathway involving the D14 receptor (and associated components) is proposed to a vascular plant innovation. The data in this manuscript goes some way in providing a unified view of the evolution of ancient signalling pathways, likely originally involved in communication between land plants and mycorrhizal fungi, being co-opted into developmental pathways in some lineages of land plants.

Overall, the manuscript is well written and easily digested, but here are a few minor comments:

Line 5 of abstract — the presence of the compound is ancestral (rather than BSB being ancestral as written)

Lines 33-35: Here it is stated that the authors confirmed the AM symbiosis — perhaps there should be a reference to earlier studies?

Line 131: delete 'extremely'

Line 139: three-week-old gemmalings ('gemmae' are not three weeks old)

Line 194 Similar, not similarly

Finally, I find the 'ligand first' or 'signalling first' proposal somewhat simplistic — does it not depend upon the perspective from which one is viewing it; or perhaps how much is known about the ancestral functions of the ligands and the pathways? The signalling pathway into which BSB was incorporated also existed prior to the assembly of the ligand dependency of the pathway, so if one was studying this pathway, the incorporation of BSB would be a later event. For example, if one discovered a function from GA-like molecules that was predicted to be present in the ancestral land plant, would that not change the perspective? For auxin, while the molecule auxin is widespread in eukaryotes (with largely unknown functions), the biosynthetic pathway present in land plants likely evolved in the ancestral land plant, whereas the transcriptional network into which auxin (and its land plant specific receptor) were integrated appears to predate land plants, and thus also be classified as a 'signalling first' mode? [See On the evolutionary origins of land plant auxin biology Cold Spring Harb Perspect Biol doi: 10.1101/cshperspect.a040048 , and references therein.]

The key question, as pointed out by the authors, is how the seemingly unconnected pathways are integrated into a single one — for this it seems that knowledge of the ancestral functions of all components is necessary, and in this respect, the BSB/SL system might provide more complete insight that is known at present for auxin and ABA. For example, it is of interest that the integration of BSB with a pre-existing signalling pathway occurred only in vascular plants, and perhaps this was selected for due to the requirement for more long distance signalling in plants the evolved roots and shoots that are spatially separated.

Reviewer #2 (Remarks to the Author):

The manuscript from Kodama et al reports the identification of a novel strigolactone (SL) molecule isolated from exudates of the liverwort *Marchantia paleacea* (belonging to the bryophytes). The authors named this SL BSB (bryosymbiol). In flowering plants, SL are known as both branching hormones and rhizospheric signals, promoting the establishment of symbiosis with arbuscular

mycorrhizal (AM) fungi. Until now, the evolutive origin of both these functions has remained elusive. Here, various elegant approaches and assays (chemistry, mutant genetics, biochemistry, cell biology) have been used to demonstrate that

- BSB is ancestral as it is found in several extant liverworts and one hornwort, but also in vascular plants;

- BSB synthesis is dependent upon CCD8 and MAX1 function (a mechanism is proposed for BSB synthesis from carlactone and carlactonoic acid);

- BSB synthesis genes are upregulated by phosphate starvation, parallel to AM symbiosis permissive state;

- BSB induces hyphal branching of AM fungus *Gigaspora margarita*;

- impaired BSB synthesis prevents *M. paleacea* to establish symbiosis with the AM fungus *Rhizophagus irregularis*,

- impaired BSB synthesis does not lead to any other visible phenotype in *M. paleacea*, making unlikely a hormonal role for this SL; MpaKAI2A and B and MpaMAX2 are not involved in BSB perception;

- introduction of the Arabidopsis SL receptor AtD14 in *M. paleacea* allows to reconstruct a SL signaling pathway in this liverwort, that relies on MpaKAI2 and MpaMAX2 endogenous functions. The MpaKAI2/MpaMAX2 pathway is BSB-independent.

These convincing results allow the authors to propose that SL ancestral role was to signal for AM symbiosis in the rhizosphere, and that SL hormonal function evolved during plant diversification following KAI2 receptors neo-functionalization. The manuscript ends with a very interesting general discussion around evolution of hormonal networks in plants.

This manuscript should interest a broad community of plant scientists, chemists and evolutionary biologists.

Please find below a list of (minor) points to improve the manuscript.

1-Fig 1c : What do Yellow, orange and red boxes refer to in the alignment?

2-Fig2a and 2b and line 106-108: explain the choice of investigated species, and label them in Suppl Fig 1 and 2

3-Fig2c: the Mpaccd8 mutant could be used as a control for this assay-also as a control for Fig5d. In addition, for these assays, a control with GR24 on the tested seeds is usually shown, as done in Fig 4d.

4-Line 78-80: NOESY correlations may not be sufficient to support the relative configuration. NMR modeling (see ref Willoughby et al, 2014 doi:10.1038/nprot.2014.042) may help to validate the structural assignments.

5-Line 90-92: « The conversion of carlactonoic acid to BSB is presumed to be via epoxidation and that upon proton abstraction of the carboxyl group, closure of the C ring occurs ». This sentence is not clear to me. Also, proton abstraction may not be necessary as acid catalysis could also lead to epoxide opening. Thus please rephrase and modulate the sentence.

6-Fig 4b: Could you please comment on why so few DEG are found in Mpaccd8 vs WT when comparing -P and +P ? See also my point 14.

7-Line 130 and Sup Table 4: The link between these data and the heatmap shown Fig 4c is not easy. The four putative transporter genes upregulated in -P could be underlined?

8-Line 132 and Fig 4d: can the germination results on Mpa exudates be compared with those using BSB fractions (Figure 2c)? 100% germination was observed with BSB fractions, but « only » 25% with Mpa exudates.

9-Fig 4e: It would be nice to have a picture for high excision zone, to compare with the picture (small) of fully colonized thalli. Please show what is the notch (mentioned in legend).

10-Fig 4f and line 140: I suggest to test MAX1 transcript levels at the 3 developmental stages, since the results on MpaCCD8B do not allow to conclude for BSB synthesis but « only » for CL.

11-Fig5a: I suggest to show quantifications of hyphal branching, with different concentrations of BSB.

12-Line 154 Fig 5c and Suppl Fig 5: Please correct, 2 double-mutants Mpaccd8a-2/b-3 and Mpaccd8a-1/b-4 are shown, and not “3 alleles” as said in the text. In addition, contrary to what is said line 158-159, Mpaccd8a-1/b-4 is not restored by GR24 addition. Please correct and temper the conclusion. It would be interesting to test BSB or GR24 separated enantiomers for this assay.

13-Line 175: Please correct bryophytes by Marchantia, since mosses are not concerned.

14-Line 182: « This is consistent with the results of the RNAseq analysis comparing gene expression between WT and Mpaccd8a/8b, which shows that only a small number of genes are differentially regulated (Fig. 4b). » Please rephrase the sentence. RNA seq comparing -P and +P in WT shows many DEG. It is only in Mpaccd8 that a small number of DEG (-P vs +P-) is observed. The shown data do not allow to conclude on DEG between WT and Mpaccd8.

15-Line 215 « as a hormone in bryophytes» : please replace bryophytes by Marchantia. In mosses, there are many KAI2 homologs and some may be BSB receptors? Also correct in the next sentence line 216.

16-Fig7a: Statistical analysis should be shown, in particular for line #7. Same for Suppl Fig 7a and Suppl Fig 7d.

17-Line 229: Please recall how both these genes are regulated (detail what you mean by « expression highly dependent on KAI2-dependent signaling »); also cite Yao et al, New Phytol 2021 for MpDLP1. In Yao et al, MpDLP1 is a marker for KL signaling, downregulated in Mpmx2 vs WT, and induced in WT by (-)-GR24 (GR24ent5DS) and not by (+)-GR24 (GR245DS).

18-Fig 7: This cross species complementation assay is very smart. I would suggest (not mandatory) to introduce a Mpmx2 mutation in the MpaAtD14ox line, to further check that signaling occurs through the KAI2-MAX2 pathway.

19-Line 245 : What do you mean by “KAI2-like signaling pathway” ? Do you mean KL (KAI2-Ligand ?) or KAI2-dependent as used above ? Line 260, you say « MpaMAX2-dependent signaling pathway », and then line 291 and 298 you say « KAI2/MAX2 signaling pathway ». All these terms refer to the same pathway I guess. Please homogenize the terms.

20-Line 344 and followings, and line 392 and followings, informations need to be added on Physcomitrium source and growth conditions, (e. g. for protonema extracts for Fig 2).

21-Line 118: Use « relative transcript levels » rather than « expression » all along the manuscript.

22-Typo on Suppl Table 4, -P-up(WT) : CCD9B.

23-Suppl Fig 6: please correct the legend, 100 microM should be between 20 and 250 microM.

24-It is preferable to use the international notation (recognized by chemists) for GR24 enantiomers, ie (+)-GR24 instead of GR245DS, (-)-GR24 instead of GR25ent-5DS, etc

25-Supp Fig6: if possible enlarge the panel and highlight the shift with (-)-GR24. Higher concentrations may be used.

26-line 312: ref 53 should not be cited here. Please cite Lopez-Obando et al 2021 that just came out <https://doi.org/10.1093/plcell/koab217>

27-line 322: please add reference to "previous studies"

28-line 409: please indicate the group of *P. ramosa* used, (group 1 or 2a depending on the field where seeds have been collected) see ref Huet et al 2020, DOI10.3389/fpls.2020.01075

Reviewer #3 (Remarks to the Author):

It is a very interesting MS. The work attempts to postulate the evolutionary motive for the synthesis of SLs by plants. They detect a new SLs in a non-vascular plant, which postulates that the first role they played in plants was as signallers and communicators with AMFs. All this is supported by genetic suppression studies to block their biosynthesis, exogenous application of SLs, etc. Nevertheless, there are some weak points of the work, from the organic chemistry point of view, related with the structural characterization of the new compound that should be addressed by the authors:

- The authors present the new compound establishing in the structure the stereochemistry of centers C-4, C-5 and C-2' but in the same work they already indicate that centers C-4 and C-5 are interchangeable. The only thing they say is that according to the NOESY correlations they must be anti each other. Therefore, to indicate the centers without stereochemistry and defined as much indicate that they must be C-4 and C-5 anti to each other, in fact, in the name of the compound they do not indicate stereochemistry of the centers. This conclusion that they are anti each other it is also not justified since the NOE effect that they say they observe is between C-5 and C-12 and C-13 methyl and between C-4 and C-14 methyl, so making this statement is not correct, at least without presenting a 3D model of the molecule, to confirm the orientation of these groups. On the other hand, they do not present the α -oriented.
- They assign the C-2' configuration by CD since they say that a negative effect is observed on 270nm, which is the same effect that is observed in other SLs. This statement should be supported by a calculation that confirms this CD. Even taking advantage of the fact that they do, they could make the calculation for all the possible configurations of the 3 centers (4R 5S 2'S, 4R 5S 2'R, 4S 5R 2'S and 4S 5R 2'R) doubtful to observe if there are significant differences between them, perhaps It is not decisive but maybe it is.

Reviewer #4 (Remarks to the Author):

This is a very interesting and original work. It has certainly potential to influence significantly further studies in the field. As far as I know this is the first report offering a model for the evolution of hormonal networks in plants. Experiments were done with care, conclusions are well supported by experimental data. As far as I can tell, methods are properly described or relevant reference is provided. I have got few minor concerns, only:

- 1) A structure elucidation of previously unidentified compounds is often confirmed by chemical synthesis. Please, describe the main obstacles in the chemical synthesis of bryosymbiol. Moreover, provide the reader with an example of another low-molecular-weight substances produced by plants in

minute quantities whose structure has been elucidated via the combination of mass spectrometric, NMR, and CD data, only.

2) line 228, Please, describe briefly the function of DLP1, so reader can estimate relevance of the marker selected.

3) Line 266 Please, rewrite the first sentence in DISCUSSION. Currently, it sounds like "SLs play a part in plant's control of AM as well as parasitism by root parasitic plants"

4) Please, make sentence line 292 understandable. It's too long and too complex. More shorter sentences does the job.

We thank all reviewers for their insightful comments and suggestions. Responses to individual comments are below.

Reviewer #1 (Remarks to the Author):

This manuscript describes the identification of a small secreted compound, BSB, whose function in *Marchantia paleacea* appears to be in a communication pathway with colonizing arbuscular mycorrhizal fungi. Its production is increased by phosphate deficiency consistent with the proposed role. As this compound is produced both in liverworts and seed plants, it appears it was produced in the ancestral land plant. The incorporation of similar molecules, i.e. strigolactones, into a signalling pathway involving the D14 receptor (and associated components) is proposed to a vascular plant innovation. The data in this manuscript goes some way in providing a unified view of the evolution of ancient signalling pathways, likely originally involved in communication between land plants and mycorrhizal fungi, being co-opted into developmental pathways in some lineages of land plants.

Overall, the manuscript is well written and easily digested, but here are a few minor comments:

Line 5 of abstract — the presence of the compound is ancestral (rather than BSB being ancestral as written)

Response: We changed the text.

Lines 33-35: Here it is stated that the authors confirmed the AM symbiosis — perhaps there should be a reference to earlier studies?

Response: Radhakrishnan *et al.* (ref 17) has been added as it demonstrates the absence of AMS in *Marchantia polymorpha*, and Humphreys *et al.* (ref 18) that demonstrated the occurrence of mutualism in *Marchantia paleacea*.

Line 131: delete ‘extremely’

Response: We have deleted ‘extremely’. (line 134)

Line 139: three-week-old gemmalings (‘gemmae’ are not three weeks old)

Response: We have changed gemmae to gemmalings. (lines 141, 143)

Line 194 Similar, not similarly

Response: This has been corrected. (line 198)

Finally, I find the ‘ligand first’ or ‘signalling first’ proposal somewhat simplistic — does it not depend upon the perspective from which one is viewing it; or perhaps how much is known about the ancestral functions of the ligands and the pathways? The signalling pathway into which BSB was incorporated also existed prior to the assembly of the ligand dependency of the pathway, so if one was studying this pathway, the incorporation of BSB would be a later event. For example, if one discovered a function from GA-like molecules that was predicted to be present in the ancestral land plant, would that not change the perspective? For auxin, while the molecule auxin is widespread in eukaryotes (with largely unknown functions), the biosynthetic pathway present in land plants likely evolved in the ancestral land plant, whereas the transcriptional network into which auxin (and its land plant specific receptor) were integrated appears to predate land plants, and thus also be classified as a ‘signalling first’ mode? [See On the evolutionary origins of land plant auxin biology Cold Spring Harb Perspect Biol doi: 10.1101/cshperspect.a040048 , and references therein.]

The key question, as pointed out by the authors, is how the seemingly unconnected pathways are integrated into a single one — for this it seems that knowledge of the ancestral functions of all components is necessary, and in this respect, the BSB/SL system might provide more complete insight that is known at present for auxin and ABA. For example, it is of interest that the integration of BSB

with a pre-existing signalling pathway occurred only in vascular plants, and perhaps this was selected for due to the requirement for more long distance signalling in plants the evolved roots and shoots that are spatially separated.

Response: We are really pleased to see that the reviewer found this section of the discussion stimulating. We fully agree with his/her comments. As he/she pointed out, our proposition of “ligand first” or “signalling first” is actually only relevant when considering a focal extant clade. We have entirely rewritten that section to make it more open and accurate (lines 328-350). We propose now a less provocative model in which the evolution of ligand/receptor links is central to the evolution of most hormonal networks found in angiosperms. Given that the case of GA is still puzzling – as we have recently commented with the loss of GA – *GID1* in bryophyte being an option (El Mahboubi & Delaux 2021) – we do not discuss it anymore in the text.

Reviewer #2 (Remarks to the Author):

The manuscript from Kodama et al reports the identification of a novel strigolactone (SL) molecule isolated from exudates of the liverwort *Marchantia paleacea* (belonging to the bryophytes). The authors named this SL BSB (bryosymbiol). In flowering plants, SL are known as both branching hormones and rhizospheric signals, promoting the establishment of symbiosis with arbuscular mycorrhizal (AM) fungi. Until now, the evolutive origin of both these functions has remained elusive. Here, various elegant approaches and assays (chemistry, mutant genetics, biochemistry, cell biology) have been used to demonstrate that

- BSB is ancestral as it is found in several extant liverworts and one hornwort, but also in vascular plants;
- BSB synthesis is dependent upon *CCD8* and *MAX1* function (a mechanism is proposed for BSB synthesis from carlactone and carlactonoic acid);
- BSB synthesis genes are upregulated by phosphate starvation, parallel to AM symbiosis permissive state;
- BSB induces hyphal branching of AM fungus *Gigaspora margarita*;
- impaired BSB synthesis prevents *M. paleacea* to establish symbiosis with the AM fungus *Rhizophagus irregularis*,
- impaired BSB synthesis does not lead to any other visible phenotype in *M. paleacea*, making unlikely a hormonal role for this SL; *MpaKAI2A* and *B* and *MpaMAX2* are not involved in BSB perception;
- introduction of the Arabidopsis SL receptor *AtD14* in *M. paleacea* allows to reconstruct a SL signaling pathway in this liverwort, that relies on *MpaKAI2* and *MpaMAX2* endogenous functions. The *MpaKAI2/MpaMAX2* pathway is BSB-independent.

These convincing results allow the authors to propose that SL ancestral role was to signal for AM symbiosis in the rhizosphere, and that SL hormonal function evolved during plant diversification following *KAI2* receptors neo-functionalization. The manuscript ends with a very interesting general discussion around evolution of hormonal networks in plants.

This manuscript should interest a broad community of plant scientists, chemists and evolutionary biologists.

Please find below a list of (minor) points to improve the manuscript.

1-Fig 1c : What do Yellow, orange and red boxes refer to in the alignment?

Response: The colored boxes refer to polymorphic amino acids. The legend has been edited to clarify this point.

“Equivalent amino acids present in one/two, three or four sequences are colored in red, yellow and orange respectively” has been added to the figure legend. (lines 663-664)

2-Fig2a and 2b and line 106-108: explain the choice of investigated species, and label them in Suppl Fig 1 and 2

Response: We analyzed nine ferns obtained from a garden store and 27 seed plants, including plants in which SLs have not been detected or analyzed so far. We have added this explanation (lines 106 to 107)

and the new Supplementary Table 6. We also labeled them (if were existed) in Supplementary Fig. 1 and 2.

3-Fig2c: the *Mpaccd8* mutant could be used as a control for this assay-also as a control for Fig5d. In addition, for these assays, a control with GR24 on the tested seeds is usually shown, as done in Fig 4d.

Response: Currently, there are not enough plant materials of the *Mpaccd8* mutants available for the bioassay. It will take several months to wait for plant growth for this. The germination activity of the BSB fraction in WT (Fig. 2c) is not present in *Mpamax1* (Fig. 5d), and the activity of the carlactone fraction of *Mpamax1* is not present in WT (no accumulation), so the respective activities are BSB and carlactone. We have indicated the germination activity of GR24 as a control in figure legends of Fig. 2c (line 677 to 678) and 5d (lines 731-732).

4-Line 78-80: NOESY correlations may not be sufficient to support the relative configuration. NMR modeling (see ref Willoughby et al, 2014 doi:10.1038/nprot.2014.042) may help to validate the structural assignments.

Response: Thank you for your suggestions to validate the structure assignments of BSB by computation of NMR chemical shifts. However, unfortunately, our group does not include experts in computational chemistry. It is well known that the determination of relative configuration of vicinally coupled protons in five-membered rings from the spin-spin coupling constants (*J* values) is very difficult since *J* values of *trans* (*anti*)- and *cis* (*syn*)-coupled protons are too close. In addition, the two sp^2 carbons in the butyrolactone ring of BSB further increase the difficulty. In the NOESY spectrum, we observed clear correlations between H-12/H-13 and H-5, and between H-14 and H-4 as shown below.

We next measured the distances between H-12/H-13/H-13 and H-4/H-5 by using Chem3D as shown below.

In the 4,5-*anti* isomer, the distances between H-12/H-13 and H-5, and between H-14 and H-4 are ca. 2 Å. By contrast, in the 4,5-*syn* isomer, the distances between H-12/H-13 and H-5 are ca. 2 Å, while that

between H-14 and H-4 is 5.6 Å that is likely to be distant enough to have no NOESY correlation between them. To confirm the stereochemistry, we attempted to recrystallize BSB for X-ray crystallography. Unfortunately, however, this SL was decomposed due to its instability. It is well known that natural SLs, especially non-canonical SLs, are highly chemically unstable. Heliolactone, a non-canonical SL isolated from sunflower, was published without determination of the stereochemistry at C-11 (Ueno et al., *Phytochemistry*, 2014). Avenaol, a non-canonical SL isolated from black oat (*Avena strigosa* Schreb.), was also published without confirmation of the stereochemistry in the A- and C-rings (Kim et al., *Phytochemistry*, 2014). We hope that the stereochemistry at C-4 and C-5 in BSB will be unambiguously determined by stereoselective total synthesis as achieved for the non-canonical SLs, heliolactone (Yoshimura et al., *Helvetica Chimica Acta*, 2019) and avenaol (Yasui, et al., *Nature communications*, 2017).

5-Line 90-92: « The conversion of carlactonoic acid to BSB is presumed to be via epoxidation and that upon proton abstraction of the carboxyl group, closure of the C ring occurs ». This sentence is not clear to me. Also, proton abstraction may not be necessary as acid catalysis could also lead to epoxide opening. Thus please rephrase and modulate the sentence.

Response: We have rephrased the sentence to “The conversion of carlactonoic acid to BSB appears to proceed via 7,8-epoxidation (carlactone numbering) followed by S_N2-type ring opening of the epoxide by the carboxyl oxygen atom to yield the possible two isomers with 4,5-*anti*-substituted butyrolactone ring.”(lines 91-94)

Yes, the epoxide ring can also be opened by acid catalysis. However, we wish to eliminate the possibility. Considering that the C-7 is allylic carbon, acid catalyzed-ring opening of the epoxide and subsequent lactone ring formation could proceed by the S_N1 mechanism to yield both 4,5-*syn* and 4,5-*anti* isomers.

6-Fig 4b: Could you please comment on why so few DEG are found in Mpaccd8 vs WT when comparing -P and +P ? See also my point 14.

Response: In Fig 4b, blue circles indicate DEGs down- or upregulated under -P condition compared to normal condition in WT. Pink circles indicate DEGs down- or upregulated in **Mpaccd8a/8b compared to WT under the normal condition (NOT DEGs responding to -P in Mpaccd8a/8b)**. We think this may be confusing and leads to a misunderstanding (such as that the pink circles show DEGs responding to -P conditions in Mpaccd8a/8b background). Therefore, we modified the legend of Fig. 4b (lines 702-706).

We think that the reason for the small number of DEGs between WT and Mpaccd8a/8b is that the products of MpaCCD8 do not affect gene expression in the *M. paleacea* cells.

7-Line 130 and Sup Table 4: The link between these data and the heatmap shown Fig 4c is not easy. The four putative transporter genes upregulated in -P could be underlined?

Response: We found that four is wrong; they are three. The three transporters are now underlined.

8-Line 132 and Fig 4d: can the germination results on Mpa exudates be compared with those using BSB fractions (Figure 2c)? 100% germination was observed with BSB fractions, but « only » 25% with Mpa exudates.

Response: Exudates used in Fig. 2c were purified with reverse-phase HPLC. In contrast, the crude culture media were used in Fig. 4d. We think that this is the main reason for the low germination frequency in Fig. 4d. We added the detailed experimental method of Fig. 4d to the Methods section (lines 432-435).

9-Fig 4e: It would be nice to have a picture for high excision zone, to compare with the picture (small) of fully colonized thalli. Please show what is the notch (mentioned in legend).

Response: We have now added two pictures (fully colonized and high exclusion zone) with the notch, and the end of the colonized zone indicated. See panel 4e. We also added detailed procedure to evaluate the colonization on low phosphate conditions (lines 518-522).

10-Fig 4f and line 140: I suggest to test MAX1 transcript levels at the 3 developmental stages, since the results on MpaCCD8B do not allow to conclude for BSB synthesis but « only » for CL.

Response: We have done qPCR analysis of MpaMAX1 expression and added the data as supplementary Fig. 5. Accordingly, Supplementary Figs 5 to 7 in the previous manuscript were changed to Figs 6 to 8.

11-Fig5a: I suggest to show quantifications of hyphal branching, with different concentrations of BSB.

Response: We evaluate the hyphal branching activity of tested compounds by determining the minimum effective concentration (MEC) using serial dilution method (Akiyama et al., Plant & Cell Physiol., 2010). To estimate MEC, we judged positive if multiple 3^o or higher order of branches appeared around the paper disk. The assay data is thus a qualitative one. BSB induces the formation of lower order branches up to the fourth order, mainly consisting of long tertiary hyphae, from the secondary hyphae of the AM fungus *Gigaspora margarita* as observed for other known natural non-canonical SLs.

12-Line 154 Fig 5c and Suppl Fig 5: Please correct, 2 double-mutants Mpaccd8a-2/b-3 and Mpaccd8a-1/b-4 are shown, and not “3 alleles” as said in the text.

Response: Thank you. This has been corrected (lines 161).

In addition, contrary to what is said line 158-159, Mpaccd8a-1/b-4 is not restored by GR24 addition. Please correct and temper the conclusion.

Response: The addition of GR24 led to most of the Mpaccd8a-1/b-4 being colonized, while no colonization at all was observed when treated with the mock solution. We rephrased the sentence to take into account the variability observed in the complementation of the Mpaccd8a-1/b-4 in terms of the number of infection points (line 162).

It would be interesting to test BSB or GR24 separated enantiomers for this assay.

Response: We agree with the reviewer that in order to determine the specificity of *R. irregularis* to different molecules, it would be interesting to test BSB and the diverse GR24 enantiomers. However, here we test whether the lack of fungal activation by strigolactones is the limiting factor in the Mpaccd8a/b mutants. As such, we reasoned that any molecule able to mimic the activity of strigolactones on *R. irregularis* would be meaningful. We designed this test based on Taulera *et al.* who have conducted similar assays in *M. truncatula* (10.1007/s00572-020-00965-9).

Given the uncoupling between the rhizospheric and the endogenous roles of strigolactones in *M. paleacea* it will be in the future an ideal model to dissect the respective contribution of the exogenous/endogenous role of strigolactones during AMS by contrasting the results with assays in angiosperms.

13-Line 175: Please correct bryophytes by Marchantia, since mosses are not concerned.

Response: This has been corrected (line 179).

14-Line 182: « This is consistent with the results of the RNAseq analysis comparing gene expression between WT and Mpaccd8a/8b, which shows that only a small number of genes are differentially regulated (Fig. 4b). » Please rephrase the sentence. RNA seq comparing -P and +P in WT shows many

DEG. It is only in *Mpaccd8* that a small number of DEG (-P vs +P-) is observed. The shown data do not allow to conclude on DEG between WT and *Mpaccd8*.

Response: This is related to comment 6. Pink circles indicate DEGs down- or up-regulated in *Mpaccd8a/8b* compared to WT under the normal condition, not DEGs (-P vs +P-) in *Mpaccd8* mutants.

15-Line 215 « as a hormone in bryophytes » : please replace bryophytes by *Marchantia*. In mosses, there are many KAI2 homologs and some may be BSB receptors? Also correct in the next sentence line 216.

Response: These have been corrected (lines 218 and 219).

We only analyzed one moss species, *P. patens*. Some of the KAI2 genes in *P. patens* may be the receptors of BSB, however, since BSB is not synthesized in *P. patens*, the KAI2s are not able to function as receptors.

16-Fig7a: Statistical analysis should be shown, in particular for line #7. Same for Suppl Fig 7a and Suppl Fig 7d.

Response: Statistic analyses have been added.

17-Line 229: Please recall how both these genes are regulated (detail what you mean by « expression highly dependent on KAI2-dependent signaling »); also cite Yao et al, *New Phytol* 2021 for MpDLP1. In Yao et al, MpDLP1 is a marker for KL signaling, downregulated in *Mpmax2* vs WT, and induced in WT by (-)-GR24 (GR24ent5DS) and not by (+)-GR24 (GR245DS).

Response: We added a description to explain how these genes were identified in our previous study (Mizuno et al., 2021) and added the reference (ref 49) as suggested (lines 230 to 236).

18-Fig 7: This cross species complementation assay is very smart. I would suggest (not mandatory) to introduce a *Mpamax2* mutation in the *MpaAtD14ox* line, to further check that signaling occurs through the KAI2-MAX2 pathway.

Response: Thank you for your suggestion. I agree with you. But we do not have the mutant lines, and it takes a long time to prepare materials due to the slow growth of *M. paleacea*. We will produce the line and use it in our future study.

19-Line 245 : What do you mean by “KAI2-like signaling pathway” ? Do you mean KL (KAI2-Ligand ?) or KAI2-dependent as used above ? Line 260, you say « *MpaMAX2*-dependent signaling pathway », and then line 291 and 298 you say « KAI2/MAX2 signaling pathway ». All these terms refer to the same pathway I guess. Please homogenize the terms.

Response: Thank you for this comment. In order to present the three layers of the hormone function, namely, ligand synthesis, perception by the receptors, and signaling, without confusion, we used ‘MAX2-dependent signaling pathway’ to mean the signaling pathway throughout the revised manuscript (lines 245-246, 251, 256, 267, 302, 306, 314).

20-Line 344 and followings, and line 392 and followings, informations need to be added on *Physcomitrium* source and growth conditions, (e. g. for protonema extracts for Fig 2).

Response: We added the information to the Methods section (lines 355 to 356, 361-362).

21-Line 118: Use « relative transcript levels » rather than « expression » all along the manuscript.

Response: We changed to relative transcription levels in Figures and text.

22-Typo on Suppl Table 4, -P-up(WT) : CCD9B.

Response: This has been corrected.

23-Suppl Fig 6: please correct the legend, 100 microM should be between 20 and 250 microM.

Response: This has been corrected. Thank you.

24-It is preferable to use the international notation (recognized by chemists) for GR24 enantiomers, ie (+)-GR24 instead of GR245DS, (-)-GR24 instead of GR25ent-5DS, etc

Response: We changed to the international notations in Figures and text.

25-Suppl Fig6: if possible enlarge the panel and highlight the shift with (-)-GR24. Higher concentrations may be used.

Response: It was difficult to enlarge the panel due to the space limitation. We made lines thinner to show the shift more clearly.

26-line 312: ref 53 should not be cited here. Please cite Lopez-Obando et al 2021 that just came out <https://doi.org/10.1093/plcell/koab217>

Response: The citation has been changed as suggested (ref 55).

27-line 322: please add reference to “previous studies”

Response: This part of the Discussion has been revised, and references have been added (line 328-).

28-line 409: please indicate the group of *P. ramosa* used, (group 1 or 2a depending on the field where seeds have been collected) see ref Huet et al 2020, DOI10.3389/fpls.2020.01075

Response: The seeds of *P. ramosa* were collected from mature plants parasitizing hemp (*Cannabis sativa*) in France, so that *P. ramosa* used in this study is group 2a. We indicated this group in Methods (lines 419-422).

Reviewer #3 (Remarks to the Author):

It is a very interesting MS. The work attempts to postulate the evolutionary motive for the synthesis of SLs by plants. They detect a new SLs in a non-vascular plant, which postulates that the first role they played in plants was as signallers and communicators with AMFs. All this is supported by genetic suppression studies to block their biosynthesis, exogenous application of SLs, etc. Nevertheless, there are some weak points of the work, from the organic chemistry point of view, related with the structural characterization of the new compound that should be addressed by the authors:

- The authors present the new compound establishing in the structure the stereochemistry of centers C-4, C-5 and C-2' but in the same work they already indicate that centers C-4 and C-5 are interchangeable. The only thing they say is that according to the NOESY correlations they must be anti each other. Therefore, to indicate the centers without stereochemistry and defined as much indicate that they must be C-4 and C-5 anti to each other, in fact, in the name of the compound they do not indicate stereochemistry of the centers. This conclusion that they are anti each other it is also not justified since the NOE effect that they say they observe is between C-5 and C-12 and C-13 methyl and between C-4 and C-14 methyl, so making this statement is not correct, at least without presenting a 3D model of the molecule, to confirm the orientation of these groups. On the other hand, they do not present the α -oriented.

Response: We are very sorry that we could not fully describe the structure determination of BSB due to the limitations of space in the main text and supplementary. To avoid confusion of readers, we modified the last sentence of the structure determination of BSB as “Although further studies are needed to confirm the absolute stereochemistry of BSB, the NOESY correlations suggested the relative

stereochemistry at C-4 and C-5 is anti and we tentatively assigned BSB to be (4*R*, 5*S*, 2'*R*) or (4*S*, 5*R*, 2'*R*).”

We proposed the **relative** configuration between H-4 and H-5 as follows. It is well known that the determination of relative configuration of vicinally coupled protons in five-membered rings from the spin-spin coupling constants (*J* values) is very difficult since *J* values of *trans* (*anti*)- and *cis* (*syn*)-coupled protons are too close. In addition, the two sp² carbons in the butyrolactone ring of BSB further increase the difficulty. In NOESY spectrum, we observed clear correlations between H-12/H-13 and H-5, and between H-14 and H-4 as shown below.

We next measured the distances between H-12/H-13/H-13 and H-4/H-5 by using Chem3D as shown below.

4,5-*anti*

4,5-*syn*

In the 4,5-*anti* isomer, the distances between H-12/H-13 and H-5, and between H-14 and H-4 are ca. 2 Å. By contrast, in the 4,5-*syn* isomer, the distances between H-12/H-13 and H-5 are ca. 2 Å, while that between H-14 and H-4 is 5.6 Å that is likely to be distant enough to have no NOESY correlation between them. To confirm the stereochemistry, we attempted to recrystallize BSB for X-ray crystallography. Unfortunately, however, this SL was decomposed due to its instability. It is well known that natural SLs, especially non-canonical SLs, are highly chemically unstable. Heliolactone, a non-canonical SL isolated from sunflower, was published without determination of the stereochemistry at C-11 (Ueno et al., *Phytochemistry*, 2014). Avenaol, a non-canonical SL isolated from black oat (*Avena strigosa* Schreb.), was also published without confirmation of the stereochemistry in the A- and C-rings. We hope that the stereochemistry at C-4 and C-5 in BSB will be unambiguously determined by stereoselective total synthesis as achieved for the non-canonical SLs, heliolactone (Yoshimura et al., *Helvetica Chimica Acta*, 2019) and avenaol (Yasui, et al., *Nature communications*, 2017).

- They assign the C-2' configuration by CD since they say that a negative effect is observed on 270nm, which is the same effect that is observed in other SLs. This statement should be supported by a calculation that confirms this CD. Even taking advantage of the fact that they do, they could make the

calculation for all the possible configurations of the 3 centers (4R 5S 2'S, 4R 5S 2'R, 4S 5R 2'S and 4S 5R 2'R) doubtful to observe if there are significant differences between them, perhaps It is not decisive but maybe it is.

Response: The assignment of the absolute configuration at C-2' in SLs have been determined by the Cotton effect around 270 nm (Frischmuth et al., *Tetrahedron Asymmetry*, 1993; Sugimoto et al., *J. Org. Chem.*, 1998; Welzel et al., *Chem. Commun.*, 1999). We also determined the absolute configuration at C-2' in BSB to be *R* according to the previous studies. Although we are not experts in computational chemistry, we attempted to elucidate the ECD of BSB isomers by using Gaussian 09 (CAM-B3LYP/6-311G(d,p)) as response to Reviewer #3. WB97XD/6-311G(d,p) gave similar spectra.

(4S,5R,2'R)

(4R,5S,2'R)

(4R,5S,2'S)

(4S,5R,2'S)

We also attempted to elucidate the ECD of 5DS and 4DO, canonical SLs, by using Gaussian 09 (CAM-B3LYP/6-311G(d,p)).

5DS (2'R)

4DO (2'R)

ent-5DS (2'S)

ent-4DO (2'S)

It is likely that these ECDs support our assignment at the C-2'. We hope that the stereochemistry of all the chiral carbons in BSB will be unambiguously confirmed by stereoselective total synthesis as achieved for the non-canonical SLs, heliolactone (Yoshimura et al., *Helvetica Chimica Acta*, 2019) and avenaol (Yasui, et al., *Nature communications*, 2017).

Reviewer #4 (Remarks to the Author):

This is a very interesting and original work. It has certainly potential to influence significantly further studies in the field. As far as I know this is the first report offering a model for the evolution of hormonal networks in plants. Experiments were done with care, conclusions are well supported by experimental data. As far as I can tell, methods are properly described or relevant reference is provided. I have got few minor concerns, only:

1) A structure elucidation of previously unidentified compounds is often confirmed by chemical synthesis. Please, describe the main obstacles in the chemical synthesis of bryosymbiol. Moreover, provide the reader with an example of another low-molecular-weight substances produced by plants in minute quantities whose structure has been elucidated via the combination of mass spectrometric, NMR, and CD data, only.

Response: Yes, we previously determined the chemical structures of some strigolactones by chemical synthesis (Seto et al., *PNAS*, 2014; Abe et al., *PNAS*, 2014; Tokunaga et al., *Phytochemistry*, 2016; Baz et al., *Molecular Plant*, 2018; Mori et al., *Phytochemistry*, 2020). Although we have also been trying to achieve the chemical synthesis of BSB, we have not yet succeeded in the total synthesis of BSB. This is because we were unable to introduce a formyl group at the alpha-carbon in the butyrolactone ring probably due to the steric hindrance caused by the adjacent beta-hydroxyl group. In fact, no reports were found on the alpha-acylation of beta-hydroxybutyrolactone from the SciFinder database.

There are many studies on the structure determination of natural products by only using MS, NMR and CD data as reported in the journal such as *Journal of Natural Products*, *Phytochemistry*, *Tetrahedron*, *Tetrahedron Letters*, and *Organic Letters* and so on.

2) line 228, Please, describe briefly the function of DLP1, so reader can estimate relevance of the marker selected.

Response: This is related to comment 17 of reviewer 2. We added a description to explain how these genes were identified in previous studies (lines 231-235) (Mizuno et al., 202 ; Yao et al 2021).

3) Line 266 Please, rewrite the first sentence in DISCUSSION. Currently, it sounds like "SLs play a part in plant's control of AM as well as parasitism by root parasitic plants"

Response: We revised this sentence and placed it at the beginning of this section (lines 274-276)

4) Please, make sentence line 292 understandable. It's too long and too complex. More shorter sentences does the job.

Response: We revised this sentence (lines 296-298) and changed the order of sentences in this paragraph.

REVIEWER COMMENTS

Reviewer #2 (Remarks to the Author):

I thank the authors for their revised version, and for addressing all my points.

I have few remarks, detailed below:

Point 1- Thank you for clarifying the legend. Still, the use of red color prevents to see the aa identity, even when zooming. I suggest to use another color.

Point 4- It is a bit disappointing that an expert in computational chemistry could not be found by the authors. Since the instability of BSB is mentioned by the authors in their response, I suggest them to report evidences of this instability in their manuscript, and to compare BSB stability with that of canonical and non-canonical SL.

Point 7- It is still not clear to me what are the evidences for pointing the three underlined genes as "putative SL transporters"? Now underlined "putative SL transporters" Fig 4c are not particularly upregulated by Phosphate starvation, compared to other genes labeled as "ABCG full length transporters". The sentence line 132-133 "ABCG transporter genes, homologs of PDR1, were upregulated under phosphate-deficient conditions" probably needs to be tempered.

Point 21- « Expression » still needs to be changed into transcript levels at several places in legends to Figure 4, Figure 6 and Figure 7, and in supplemental figures.

Reviewer #4 (Remarks to the Author):

The following explanation should be integrated into the main text of the article. Explanation to the reviewer is one thing, but a reader of the article needs to be informed. I recommend avoiding mentioning SciFinder database, I would prefer universal "in literature".

"Although we have also been trying to achieve the chemical synthesis of BSB, we have not yet succeeded in the total synthesis of BSB. This is because we were unable to introduce a formyl group at the alpha-carbon in the butyrolactone ring probably due to the steric hindrance caused by the adjacent beta-hydroxyl group. In fact, no reports were found on the alpha-acylation of beta-hydroxybutyrolactone from the SciFinder database."

Reviewer #5 (Remarks to the Author):

In the present form, the conclusion reached by the Authors seems - in my opinion - not fully supported by the NMR experiments realized. This means - under no circumstance - that the result is not correct but the rationale behind it is not evident.

Some points merit mention:

1) I was not able to find NMR data - a part of a summary description at pp 23-24 of the manuscript - as well as it seems in the version I received no NMR chart is supplied. Please note this is rather unusual nowadays: at least ¹H and ¹³C-NMR are provided to the journal and, considering the complexity, also 2D-NMR experiments should be provided and commented in detail.

2) To solve the stereochemical issue with the aim to unambiguously determine the configuration, I would recommend to try a X-ray structural analysis in case of availability of small amounts of material. Also, in this sense, I would stress that pure chemical aspects (physical state, sequence adopted for isolation) on the substance are missing.

3) All the chemical description of bryosymbiol should be collected in a dedicated paragraph of the Supp. Information to have a clear focus on its chemistry and structural elucidation.

Reviewer #2 (Remarks to the Author):

I thank the authors for their revised version, and for addressing all my points. I have few remarks, detailed below:

Point 1- Thank you for clarifying the legend. Still, the use of red color prevents to see the aa identity, even when zooming. I suggest to use another color.

Response: We changed the color to green.

Point 4- It is a bit disappointing that an expert in computational chemistry could not be found by the authors. Since the instability of BSB is mentioned by the authors in their response, I suggest them to report evidences of this instability in their manuscript, and to compare BSB stability with that of canonical and non-canonical SL.

Response: We successfully determined the unambiguous chemical structure of bryosymbiol using computational chemistry, revised that section on structure determination in the text (Line 71 to 75 and 465 to 499), and added new data as supplementary Figures 3-8 and supplementary Data.

Regarding the stability of bryosymbiol, it is impossible to quantify bryosymbiol due to the lack of a synthetic sample. Therefore, we cannot compare the stability of bryosymbiol with that of known strigolactones.

Point 7- It is still not clear to me what are the evidences for pointing the three underlined genes as "putative SL transporters"? Now underlined "putative SL transporters" Fig 4c are not particularly upregulated by Phosphate starvation, compared to other genes labeled as "ABCG full length transporters". The sentence line 132-133 "ABCG transporter genes, homologs of PDR1, were upregulated under phosphate-deficient conditions" probably needs to be tempered.

Response: We agree with the reviewer. The underlined genes are DEGs upregulated under -P condition. Therefore, we changed the figure legend accordingly.

Point 21- « Expression » still needs to be changed into transcript levels at several places in legends to Figure 4, Figure 6 and Figure 7, and in supplemental figures.

Response: We changed all <expression> in legends of Fig 4, 6, and 7 and SFig. 5, and 8 to <transcript level>.

Reviewer #4 (Remarks to the Author):

The following explanation should be integrated into the main text of the article. Explanation to the reviewer is one thing, but a reader of the article needs to be informed. I recommend avoiding mentioning SciFinder database, I would prefer universal "in literature".

"Although we have also been trying to achieve the chemical synthesis of BSB, we have not yet succeeded in the total synthesis of BSB. This is because we were unable to introduce a formyl group at the alpha-carbon in the butyrolactone ring probably due to the steric hindrance caused by the adjacent beta-hydroxyl group. In fact, no reports were found on the alpha-acylation of beta-hydroxybutyrolactone from the SciFinder database."

Response: Thank you for pointing this out. We added 'Structure confirmation by chemical synthesis also failed due to our inability to introduce a formyl group at the alpha-carbon in the butyrolactone ring. In fact, no reports were found on the α -acylation of β -hydroxybutyrolactone in literature.' in the text of Methods (Line 497 to 499)

Reviewer #5 (Remarks to the Author):

In the present form, the conclusion reached by the Authors seems - in my opinion - not fully supported by the NMR experiments realized. This means - under no circumstance - that the result is not correct but the rationale behind it is not evident.

Some points merit mention:

1) I was not able to find NMR data - a part of a summary description at pp 23-24 of the manuscript - as well as it seems in the version I received no NMR chart is supplied. Please note this is rather unusual nowadays: at least ^1H and ^{13}C -NMR are provided to the journal and, considering the complexity, also 2D-NMR experiments should be provided and commented in detail.

Response: NMR chart was presented as supplementary figures in the previously submitted manuscript. The figures might be small and insufficient. Therefore, in this revised manuscript, we complied all NMR data as supplementary Data. As for the 2D-NMR analysis, we present the minimum description as the text (Line 464 to 498 in the highlighted text) and provide detailed information and data as supplementary Figures 3-8 and supplementary Data.

2) To solve the stereochemical issue with the aim to unambiguously determine the configuration, I would recommend to try a X-ray structural analysis in case of availability of small amounts of material. Also, in this sense, I would stress that pure chemical aspects (physical state, sequence adopted for isolation) on the substance are missing.

Response: We successfully determined the unambiguous chemical structure of bryosymbiol using computational chemistry, revised that section on structure determination in the text (Line 71 to 75 and 465 to 499), and added new data as supplementary Figures 3-8 and supplementary Data.

We also added the sentence 'In an attempt to further confirm the stereochemistry by X-ray crystallography, this compound was decomposed due to its instability during recrystallization.' in the Methods section (Line 496 to 497).

3) All the chemical description of bryosymbiol should be collected in a dedicated paragraph of the Supp. Information to have a clear focus on its chemistry and structural elucidation.

Response: Thank you for this suggestion. We present all the chemical descriptions of bryosymbiol in supplementary Figures 3-8 and one of the files as supplementary Data.

REVIEWERS' COMMENTS

Reviewer #5 (Remarks to the Author):

The comments raised by the Reviewers have been adequately considered by the Authors and, in my opinion, the manuscript can be accepted in the present form.